# Heat shock protein DNAJA2 controls insulin signaling and glucose homeostasis by preventing spontaneous insulin receptor endocytosis

Yuanhua Qin[1,2,6], Wenjun Wu[1,6], Kequan Lin[3], Anthony J. Davis [4] & Yaping Huang [1,2,5] ✉

Dysregulation of heat shock protein DNAJA2 induces genomic instability and was consequently hypothesized to promote tumorigenesis. However, *DNAJA2* knockout mice do not develop cancer but exhibit neonatal lethality and the underlying mechanism remains unknown. Here, we demonstrate that DNAJA2 maintains homeostatic glucose metabolism by regulating insulin signaling. Mechanistically, DNAJA2 binds to the insulin receptor (IR) and prevents adaptor protein 2 (AP2)-mediated spontaneous IR endocytosis by inhibiting the IR-AP2 interaction. Thus, DNAJA2 defects lead to reduced IR localization on the plasma membrane and suppression of the insulin-stimulated signaling cascade, thereby inhibiting glycogen synthesis and storage in the liver during embryogenesis, further resulting in neonatal lethality of DNAJA2-deficient mice. Analysis of public datasets reveals a strong association between DNAJA2 and metabolic phenotypes, including type 2 diabetes mellitus (T2DM) and obesity, in both humans and mice. In conclusion, our study elucidates the mechanism by which DNAJA2 regulates IR endocytosis, insulin signaling and glucose metabolism, shedding light on the pathogenesis of metabolic disorders.

Heat shock proteins (HSPs) play fundamental roles in maintaining proteostasis and cellular homeostasis[1]. Dysregulation of HSPs is implicated in various human diseases, including cancer and metabolic disorders[2–6]. Defect of DNAJA2, a member of the DNAJA subfamily HSP40, promotes genome stability in mammalian cells by regulating transcription-coupled nucleotide excision repair (TC-NER) and chromosomal segregation during mitosis[7,8]. Genomic instability is typically associated with spontaneous tumorigenesis in animal models and humans[9]. Surprisingly, DNAJA2 whole-body knockout mice do not develop cancer, but exhibit neonatal lethality (incomplete penetrance)[10,11] without obvious defects in organ morphology. The underlying mechanisms responsible for the unexpected phenotype remains elusive.

Insulin regulates systemic metabolism in mammals, playing crucial roles in tissue growth, organ development, and overall survival. The insulin-driven signaling pathway in the liver is particularly important for controlling glucose and lipid metabolism[12,13]. Insulin resistance disrupts metabolic homeostasis and leads to systemic metabolic disorders in humans, including type 2 diabetes mellitus (T2DM) and obesity[14]. Insulin signaling is initiated by the binding of

[1]Institue for Molecular and Cellular Therapeutics, Chinese Institutes for Medical Research, Beijing, China. [2]School of Basic Medical Sciences, Capital Medical University, Beijing, China. [3]Department of Cardiology of the Second Affiliated Hospital, Zhejiang University School of Medicine, Hangzhou, China. [4]Department of Radiation Oncology, University of Texas Southwestern Medical Center, Dallas, TX, USA. [5]Beijing Shijitan Hospital of Capital Medical University, Beijing, China. [6]These authors contributed equally: Yuanhua Qin, Wenjun Wu. ✉e-mail: huangyp@cimrbj.ac.cn

insulin to the insulin receptor (IR) on the plasma membrane[12]. This binding induces autophosphorylation of the IR, which in turn activates downstream signaling cascades by phosphorylating numerous proteins including, insulin receptor substrate 1/2 (IRS1/2), PI3K, and AKT. This cascade ultimately activates terminal effector proteins, such as glycogen synthase kinase-3β (GSK3β), promoting cell proliferation and growth by regulating glucose uptake, glycogen synthesis, gluconeogenesis, and lipid and protein synthesis[12]. Despite the foundational understanding of the insulin signaling pathway, its complex regulations remain unclear.

HSP70 proteins modulate insulin signaling in response to various cellular stresses and are associated with the pathophysiology of insulin resistance and T2DM[4,5,15]. Recent clinical evidence and genetic studies in mice indicate that several HSP40 members are linked to insulin resistance, glucose metabolism, and T2DM[16], although the underlying mechanisms remain undetermined. Given that the glucose homeostasis of newborn mice is critical for maintaining energy homeostasis and survival before they can obtain milk[17], we hypothesize that DNAJA2 regulates postnatal survival by controlling insulin signaling and glucose homeostasis.

Here, we report that DNAJA2 deficiency impairs liver glycogen content and disrupts glucose homeostasis in newborn mice, contributing to the neonatal lethality observed in whole-body *DNAJA2* knockout mice. Consistently, liver-specific ablation of DNAJA2 induces systemic glucose intolerance and insulin resistance in mice, suggesting that DNAJA2 plays a critical role in maintaining glucose metabolism homeostasis. Mechanistic studies reveal that DNAJA2 regulates the insulin signaling pathway by counteracting spontaneous IR endocytosis. DNAJA2 interacts with IR on the plasma membrane and inhibits adaptor protein 2 (AP2)-mediated spontaneous endocytosis of IR. The absence of DNAJA2 significantly reduces IR's plasma membrane localization, causes insulin resistance and disrupts glucose metabolism and homeostasis. Analysis of published datasets in humans and mice also reveals that DNAJA2 dysregulation is associated with abnormal glucose metabolic phenotypes, including T2DM and obesity. Therefore, our study elucidates the mechanism by which DNAJA2 regulates IR endocytosis, insulin signaling, and glucose metabolism, and highlights targets for T2DM predisposition and intervention.

## Results
### DNAJA2 depletion causes liver glycogen shortage and postnatal lethality in mice
The observation that DNAJA2 deficiency induces genomic instability[7,8] prompted us to investigate its role in spontaneous tumorigenesis in mouse models. To this end, we generated *DNAJA2* homozygous knockout (*DJ2⁻/⁻*) mice by breeding heterozygous (*DJ2⁺/⁻*) pairs. The *DJ2⁻/⁻* mice were born at the expected Mendelian ratio. No significant differences were observed between wild-type (WT) and *DJ2⁺/⁻* littermates; however, *DJ2⁻/⁻* newborn pups and E18.5 embryos showed mild growth retardation compared to their WT counterparts (Fig. 1A and Supplementary Fig. 1A). The majority of *DJ2⁻/⁻* newborns died within 24 h of birth, with approximately 5−10% surviving to adulthood (Supplementary Fig. 1B), consistent with recently published studies[10,11]. The surviving *DJ2⁻/⁻* mice continued to exhibit mild growth retardation into adulthood (Supplementary Fig. 1C−D). These data indicate that DNAJA2 ablation causes neonatal lethality (incomplete penetrance) and growth defects.

To identify potential defects contributing to neonatal lethality, we compared the major organs of WT and *DJ2⁻/⁻* littermates. Morphological and development defects were not observed in key organs, including the heart, lung, and liver. However, histological analysis of liver sections revealed a significant decrease in cytoplasmic vacuolation in hepatocytes from *DJ2⁻/⁻* newborns and E18.5 embryos compared to their WT and *DJ2⁺/⁻* littermates (Fig. 1B and Supplementary Fig. 1E), implying that DNAJA2-deficiency leads to glycogen storage defects in

the liver. To test this possibility, we performed periodic acid Schiff (PAS) staining of liver sections to measure the level of glycogen, a type of polysaccharides. We observed significantly reduced PAS staining in the liver sections of *DJ2⁻/⁻* newborns and E18.5 embryos compared to those of WT littermates (Fig. 1B and Supplementary Fig. 1E), indicating that DNAJA2 depletion causes reduced glycogen storage in the liver. This is confirmed when we directly measured and compared the glycogen level in the livers between WT and *DJ2⁻/⁻* newborns (Fig. 1C).

Given that stored glycogen in the liver is the primary glucose resource for newborn mice to maintain energy homeostasis prior to nursing[17], we hypothesized that the neonatal lethality in *DJ2⁻/⁻* mice is caused by low blood sugar, a symptom called hypoglycemia. Supporting this hypothesis, we observed lower blood glucose levels in starved *DJ2⁻/⁻* newborns compared to their WT littermates (Supplementary Fig. 2A). To test whether the observed hypoglycemia leads to the neonatal lethality in *DJ2⁻/⁻* mice, we performed a glucose rescue experiment by subcutaneously injecting glucose into the newborn pups within the first 24 h post-birth. The results demonstrated that glucose injection significantly improved the survival rate of *DJ2⁻/⁻* newborns (Supplementary Fig. 2B). Therefore, our data indicate that insufficient glycogen storage in livers contributes to *DJ2⁻/⁻* neonatal lethality.

Liver glycogen is primarily synthesized from the serum glucose taken by hepatocytes in response to insulin stimulation[12]. The reduced glycogen levels observed in *DJ2⁻/⁻* newborn livers led us to investigate if the inefficiently utilized glucose remains in the blood, leading to hyperglycemia in *DJ2⁻/⁻* embryos. Indeed, *DJ2⁻/⁻* embryos exhibited higher blood glucose levels when the dam was in a feeding state (Supplementary Fig. 2C), indicating disrupted glucose metabolism and homeostasis. Biochemical analysis of serum insulin levels revealed hyperinsulinemia in *DJ2⁻/⁻* embryos (Supplementary Fig. 2D). Glucose tolerance assay, which determines how sugar is removed from the blood into other tissues, further confirmed that *DJ2⁻/⁻* embryos were much less tolerant to glucose injection than WT embryos (Supplementary Fig. 2E). Collectively, these findings indicate that DNAJA2 is necessary for maintaining glucose homeostasis and viability of newborn mice.

### Liver-specific DNAJA2 ablation disrupts the homeostasis of glucose metabolism
The liver plays a pivotal role in glucose metabolism, and insulin resistance in this organ can lead to severe metabolic abnormalities. To elucidate the role(s) of DNAJA2 in liver glucose metabolism, we generated liver-specific *DJ2⁻/⁻* mouse (CKO) via crossing *DJ2*ᶠˡᵒˣ/ᶠˡᵒˣ mice with *Albumin-Cre* mice. We observed that CKO mice were born at the normal Mendelian ratio, survived to adulthood (Supplementary Fig. 2F), and were morphologically indistinguishable from WT mice (Supplementary Fig. 2G, H). Histological analyses indicated normal liver development in the CKO mice (Supplementary Fig. 2I). However, hepatic glycogen levels in CKO mice were reduced in comparison with those in WT animals (Fig. 1D and Supplementary Fig. 2I), resembling the phenomenon observed in whole-body *DJ2⁻/⁻* mice (Fig. 1B, C). To assess whether liver-specific depletion of DNAJA2 induces glucose metabolic defects, we compared blood glucose and serum insulin levels between WT and CKO adult mice in the feeding state. We observed higher blood glucose and serum insulin in CKO mice than WT littermates (Fig. 1E, F), indicating that liver-specific DNAJA2 deletion leads to systemic glucose metabolic abnormalities. Glucose tolerance and insulin tolerance tests further confirmed that *DNAJA2* deficiency in the liver disrupts glucose metabolic homeostasis (Fig. 1G, H). To test if DNAJA2 deficiency impairs hepatocyte function, we measured glycogen synthase (GS) activity upon insulin stimulation in livers of WT and CKO mice. As shown in Fig. 1I, the GS activity is significantly reduced in CKO mice compared to their WT littermates.

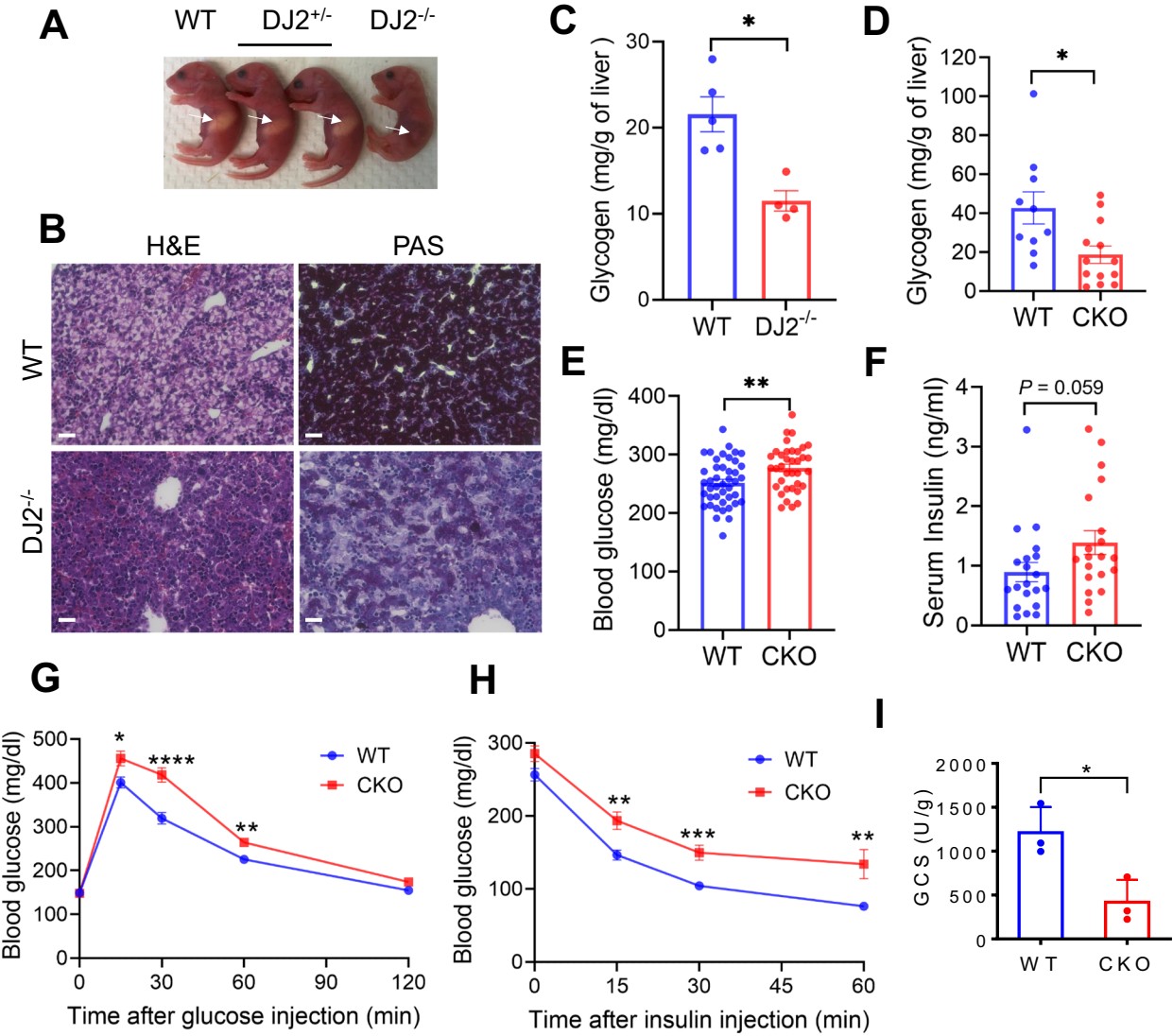

**Fig. 1 | DNAJA2 deficiency disrupts the homoestasis of glucose metabolism in mice. A** Morphologies of wild-type (WT), *DNAJA2* heterozygous (DJ2⁺/⁻) and knockout (DJ2⁻/⁻) littermates at birth (P0). Arrows indicate milk spots.
**B** Representative images of H&E staining and periodic acid-Schiff (PAS) staining of livers from WT and DJ2⁻/⁻ newborns. Scale bar, 100 μm. n = 2 experimental repeats. **C** Liver glycogen levels in WT (n = 5) and DJ2⁻/⁻ (n = 4) newborns. **D** Liver glycogen levels in 2-month-old WT (n = 10) and liver-specific *DNAJA2* knockout (CKO) mice (n = 13). **E** Fed blood glucose levels of WT (n = 43) and CKO mice (n = 36) at 2-month-

old. **F** Serum insulin levels in WT and CKO mice at 2-month-old (n = 20). Glucose tolerance (**G**) and insulin tolerance (**H**) tests in 2-month-old WT (n = 25, 18) and CKO mice (n = 22, 14). **I** Glycogen synthase activity of WT and CKO livers treated with insulin (1 U/kg body weight) for 10 min (n = 3). Data are shown as means ± SEM. P values were determined by two-tailed unpaired t test with Welch's correction. *$p < 0.05$; **$p < 0.01$; ***$p < 0.001$, ****$p < 0.0001$. Source data are provided as a Source Data file.

## DNAJA2-deficiency impairs insulin signaling

Since the insulin signaling cascade is the principal pathway governing glucose uptake and utilization, including glycogen synthase-mediated glycogen synthesis, in hepatocytes, we postulated that DNAJA2 might regulate the insulin signaling pathway. To test this, we initially compared the level of insulin-stimulated autophosphorylation of IRβ (pIRβ) in liver lysates derived from WT and *DJ2⁻/⁻* pups after a 4 h starvation period. The results revealed a significant decrease in pIRβ in DNAJA2-depleted pups compared to WT controls (Fig. 2A, B), indicating that DNAJA2 modulates insulin signaling in mouse livers. To further validate this observation and elucidate how DNAJA2 regulates insulin signaling, we administered insulin through intraperitoneal injection to both WT and CKO mice. Liver samples were collected 10 min post-injection for immunoblot analysis of pIRβ, phosphorylation of the downstream target AKT (pAKT) and the effector protein GSK3β (pGSK3β). As depicted in Fig. 2C, D, while insulin treatment

robustly induced activation of the insulin phosphorylation cascade (pIRβ, pAKT and pGSK3β) in WT livers, this response was significantly attenuated in DNAJA2-depleted CKO livers. These results strongly suggest that DNAJA2 is essential for insulin signaling.

To confirm that DNAJA2 regulates insulin signaling in vitro, we isolated hepatocytes from WT and *DJ2⁻/⁻* mice and treated them with insulin, then assessed the insulin signaling cascade via immunoblotting. Consistently, the data revealed that pIRβ and pGSK3β were attenuated in DJ2⁻/⁻ hepatocytes compared to WT cells (Fig. 2E). Additionally, we observed a significant decrease in insulin signaling in the liver cancer cell line HepG2 following knockdown of DNAJA2 (SiDJ2) (Fig. 2F, G), and this effect was consistent when DNAJA2 was knocked out in the mouse syngeneic cancer cell lines 4T1 and MC38 (Supplementary Fig. 3A–C). Pan-phosphatase inhibitor treatment could not restore the insulin-stimulated pIRβ in DJ2⁻/⁻ cells (Supplementary Fig. 3D). These findings collectively demonstrate that DNAJA2

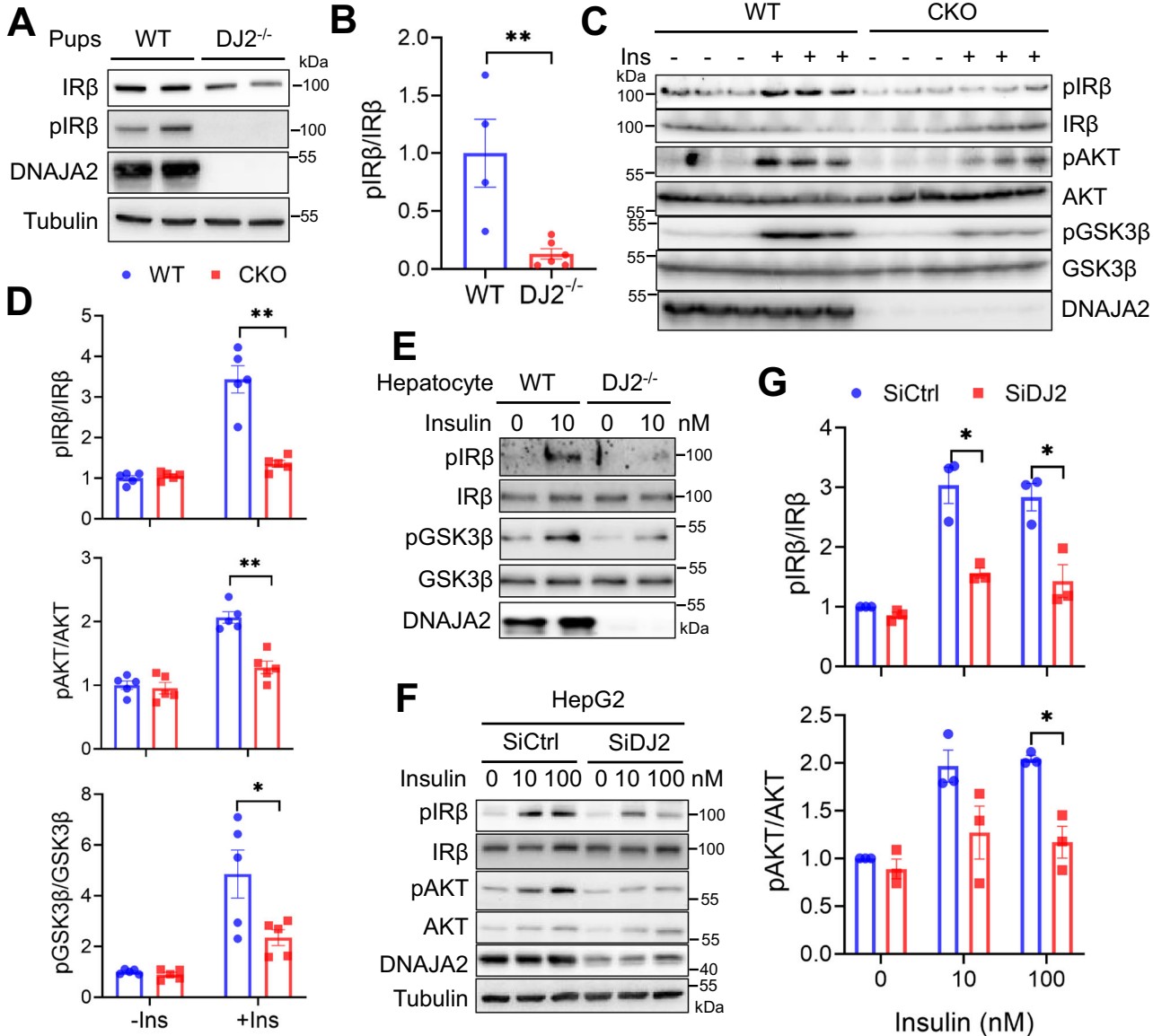

**Fig. 2 | DNAJA2 deficiency impairs insulin signaling. A** Immunoblot analyses of phosphorylated IRβ (pIRβ) in liver lysates from WT and DJ2[-/-] newborns starved for 4 h (h). **B** Quantification of the normalized pIRβ levels ((pIRβ/IRβ) in WT and DJ2[-/-] livers, as shown in (**A**). **C** Immunoblot analyses of pIRβ, phosphorylated AKT (pAKT) and GSK3β Ser9 (pGSK3β) in liver lysates from WT and CKO mice treated with (+) or without (−) insulin (Ins) at 1 U/kg body weight for 10 min. **D** Quantifications of relative levels of pIRβ, pAKT and pGSK3β shown in (**C**). **E** Immunoblots of pIRβ and

pGSK3β from WT and DJ2[-/-] primary hepatocytes stimulated with 10 nM insulin for 20 min. Immunoblots (**F**) and quantifications (**G**) of pIRβ and pAKT in control (SiCtrl) and *DNAJA2* knockdown (SiDJ2) HepG2 cells treated with the indicated concentrations of insulin for 20 min. Data are shown as means ± SEM. P values were determined by two-tailed unpaired t test with Welch's correction (**B**, **D**) or paired t test (**G**). *$p < 0.05$; **$p < 0.01$. Source data are provided as a Source Data file.

directly regulates the insulin signaling pathway at the IR node without affecting its dephosphorylation.

## DNAJA2 inhibits spontaneous clathrin-mediated endocytosis of IR

Insulin treatment activates the IR and triggers IR internalization to initiate downstream cascades through endocytosis[12]. However, spontaneous endocytosis of the IR under basal conditions results in defective plasma membrane (PM) localization of the IR, thereby impairing insulin signaling activation at the IR node upon insulin stimulation[18–20]. To ascertain whether DNAJA2 regulates insulin signaling by influencing IR PM localization, we performed immuno-fluorescence analysis of IR in control (SiCtrl) and *DNAJA2* knockdown (SiDJ2) HepG2 cells expressing GFP-IR after 15 h of serum-starvation to eliminate the influence of serum insulin-induced IR endocytosis.

As illustrated in Fig. 3A, B, the majority of IR was located to the PM in SiCtrl cells, whereas PM IR translocated to the intracellular com-partment and colocalized with RAB7, a maker of late endosomes, in SiDJ2 cells. This observation was similarly noted in 4T1 and HeLa cells (Supplementary Fig. 4A, B). Membrane fractionation assay also showed reduced IR protein level in the membrane fraction of DJ2[-/-] cells when compared to WT cells (Fig. 3C, D), suggesting that DNAJA2-deficiency induces spontaneous IR internalization through endocytosis. Since late endosome transports cargos to lysosome for degradation, we postulated that DNAJA2 deficiency promotes IR degradation. Consistent with this hypothesis, we found that the IR protein was degraded faster in DJ2[-/-] cells than WT cells (Supple-mentary Fig. 4C, D).

IR can undergo internalization through both clathrin-dependent and -independent endocytosis pathways[21]. To determine which

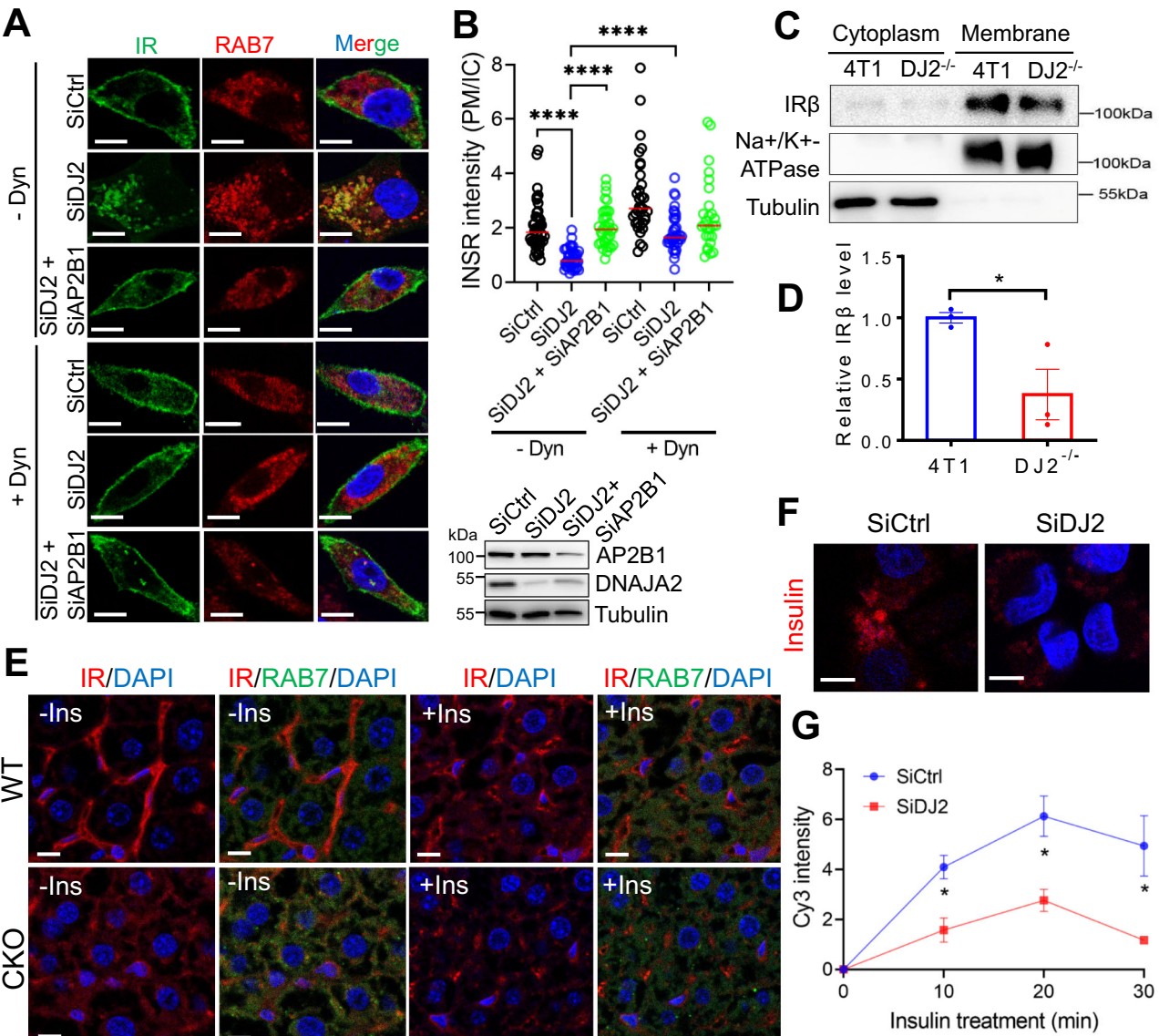

**Fig. 3 | DNAJA2 inhibits spontaneous IR endocytosis under basal state.**
**A** immunofluorescence (IF) analysis of GFP-tagged IR (green) and RAB7 (red) in HepG2 cells transfected with various siRNAs or treated with dynasore (Dyn) as indicated. HepG2 cells stably expressing GFP-IR were transfected with the indicated siRNAs for 48 h, serum strarved for 15 h and then treated with or without 50 μM dynasore for 4 h, followed by IF analysis. **B** Quantifications of the ratios of IR localized on the plasma membrane (PM) to those in intracellular compartment (IC). **C** Western blot analysis of IRβ protein levels in the membrane fraction of WT and DJ2−/− 4T1 cells. **D** Quantifications of relative IRβ levels in membrane fractions as shown in **C**. **E** Representative images showing IR, RAB7 and DAPI staining in liver sections of 2-month-old WT and CKO mice injected with (+Ins) or without (−Ins) insulin at 1 U/kg body weight for 10 min. **F** Representative images showing the intensities of endocytosed Cy3-insulin after 20 min treatment in SiCtrl and SiDJ2 HepG2 cells. **G** Quantifications of the intensities of endocytosed Cy3-insulin at different time points. Scale bar, 10 μm. Data are shown as means ± SEM. P values were determined by two-tailed unpaired t test with Welch's correction (**B**) or paired t test (**D**, **G**). *$p < 0.05$; ****$p < 0.0001$. Source data are provided as a Source Data file.

pathway is affected by DNAJA2 loss, we analyzed the subcellular localization of IR in SiDJ2 HepG2 cells treated with Dynasore[22], an inhibitor of dynamin, which is essential for clathrin-mediated endocytosis (CME). The results revealed that dynamin inhibition restored the PM localization of IR (Fig. 3A, B and Supplementary Fig. 4A). Similar findings were observed upon knockdown of the CME component adaptor protein 2 (AP2) subunit beta 1 (AP2B1)[23] using siRNAs (Fig. 3A, B), indicating that DNAJA2 prevents the spontaneous internalization of IR through the CME pathway.

To investigate whether DNAJA2 regulates IR internalization in vivo, we conducted immunohistochemistry-immunofluorescence experiments to detect IR subcellular localization in mouse liver sections with or without insulin treatment. As depicted in Fig. 3E and Supplementary Fig. 4E, the majority of IR was located on the PM in WT

livers before insulin stimulation, with internalization occurring upon insulin treatment. However, in DNAJA2-deficient livers, the intensity of IR on the PM was notably lower, and internalized IR colocalized with RAB7 even in the absence of insulin treatment.

IR activation is followed by IR internalization together with the insulin bound to it. To determine if DNAJA2 deficiency impairs insulin endocytosis, we performed a time course analysis to assess the intracellular intensity of Cy3-labled insulin in SiCtrl and SiDJ2 HepG2 cells after treating the cells with Cy3-insulin. In parallel, Alexa 568-conguated transferrin, another client of CME, was used as a control. As shown in Fig. 3F, G and Supplementary Fig. 4F, G, while the internalized transferrin intensities were essentially the same between the two cell lines, the endocytosed insulin intensity was significantly lower in DNAJA2-depleted cells compared to the control cells, indicating that

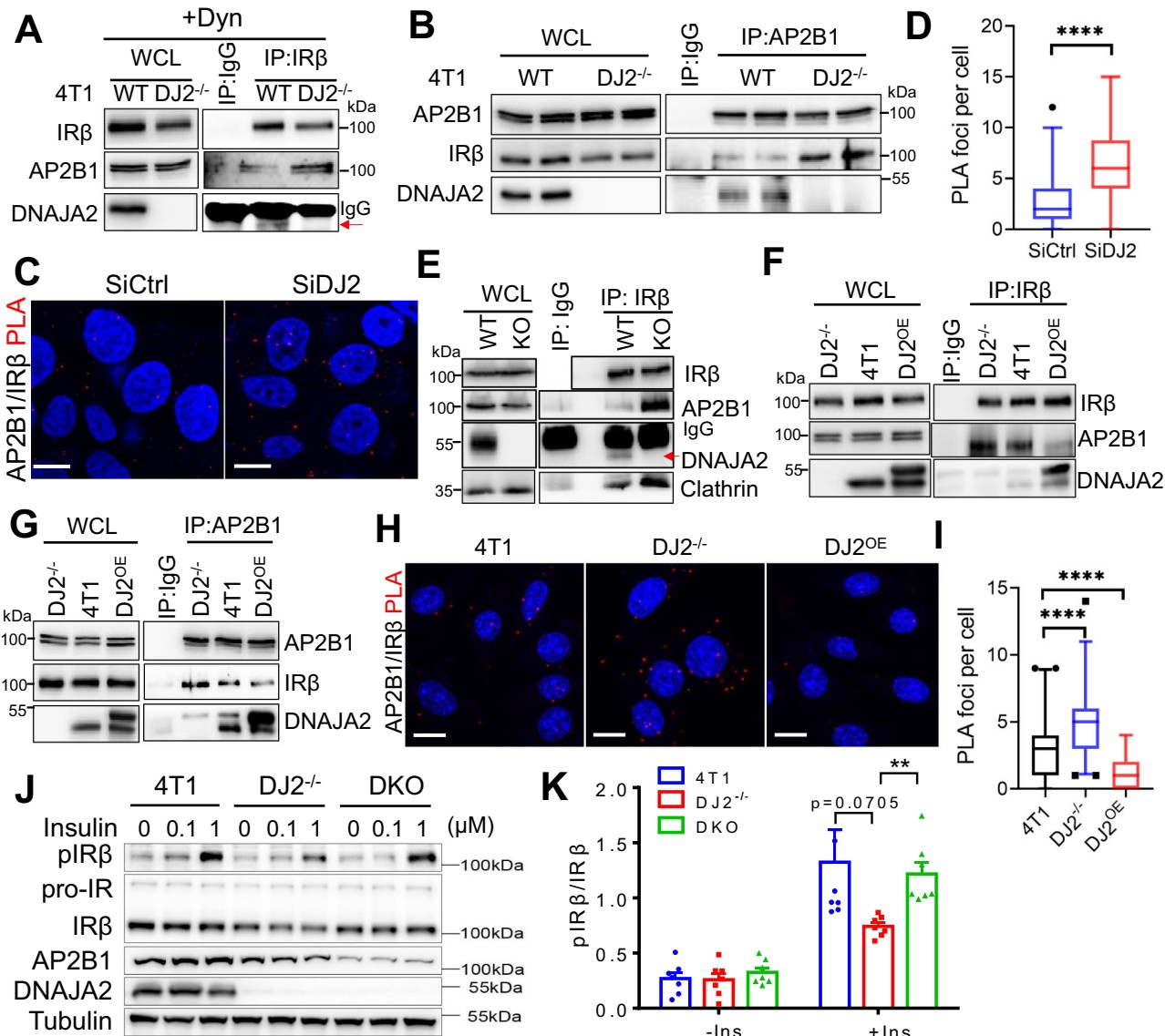

**Fig. 4 | DNAJA2 regulates insulin signaling by counteracting AP2-IR interaction.**
**A** Co-immunoprecipitation (co-IP) assay showing the interaction between IRβ and AP2B1 in WT and DJ2−/− 4T1 cells treated with 50 μM dynasore for 4 h. The red arrow indicates DNAJA2 band. **B** A reverse co-IP assay using an anti-AP2B1 antibody shows the interaction between IRβ and AP2B1 in WT and DJ2−/− 4T1 cells. Two replicates were immunoblotted for each cell line. **C** Proximity ligation assay (PLA) showing the interaction between IRβ and AP2B1 in SiCtrl and SiDJ2 HepG2 cells. **D** Quantification of the PLA foci per cell as shown in (**C**). **E** Co-IP assay showing the interaction between IRβ and AP2B1 in WT and DJ2−/− (KO) liver lysates. The red arrow indicates DNAJA2 band. **F**, **G** Co-IP assays showing the interaction between IRβ and AP2B1 in WT, DJ2−/− and DNAJA2-overexpressed (DJ2OE) 4T1 cells. An anti-IRβ (**F**) or anti-AP2B1 antibody was used in the IP experiments. Representative images (**H**) and quantifications (**I**) of the PLA assay showing the interactions between IRβ and AP2B1 in WT, DJ2−/− and DJ2OE 4T1 cells. **J** Immunoblot assay showing phosphorylated IRβ (pIRβ) in WT, DJ2−/− and *DNAJA2/AP2B1* double KO (DKO) 4T1 cells treated with the indicated concentrations of insulin for 20 min. **K** Quantifications of the realtive levels of pIRβ as shown in (**J**). Scale bar, 10 μm. Data are shown as median (**D**, **I**) or means ± SEM (**K**). P values were determined by two-tailed unpaired t test with Welch's correction. ****p < 0.0001. Source data are provided as a Source Data file.

DNAJA2 specifically affects insulin endocytosis. Overall, these results demonstrate that DNAJA2 inhibits spontaneous IR endocytosis mediated by the CME pathway.

**DNAJA2 regulates insulin signaling by counteracting the AP2-IR interaction to block IR endocytosis**

The role of DNAJA2 in preventing IR endocytosis led us to investigate whether DNAJA2 binds to IR. We performed co-immunoprecipitation (co-IP) experiments using an IRβ antibody in various cell lines, both in the presence or absence of insulin treatment. DNAJA2 interacted with IRβ in all tested cell lines, irrespective of insulin stimulation (Supplementary Fig. 5A–C). This finding was further validated in HepG2 cells using a proximity ligation assay (PLA) to quantify the DNAJA2-IRβ

interaction (Supplementary Fig. 5D, E), indicating that DNAJA2 binds to IRβ under basal conditions.

The AP2 complex plays a crucial role in clathrin-mediated endocytosis by interacting with both clathrin and cargo[23]. We hypothesized that the binding of DNAJA2 to IR might suppress the interaction between AP2 and IR, thereby inhibiting IR internalization. To test this hypothesis, we performed co-IP experiments using an IRβ antibody in WT and DJ2−/− 4T1 cells treated with Dynasore. As shown in Fig. 4A, the interaction between IRβ and AP2B1, a subunit of the AP2 heterotetramer, significantly increased in the absence of DNAJA2. Similar results were obtained in a reverse co-IP experiment using an AP2B1 antibody (Fig. 4B). The interaction between AP2B1 and IRβ was further confirmed in SiCtrl and SiDJ2 HepG2 cells using the PLA assay (Fig. 4C,

D). Additionally, the in vivo interaction between AP2B1 and IRβ was significantly enhanced in the DJ2$^{-/-}$ liver tissue compared to the WT tissue (Fig. 4E). These results collectively suggest that DNAJA2 and AP2 compete with each other for IR binding, indicating a potential mechanism by which DNAJA2 regulates IR endocytosis. To further investigate this possibility, we performed co-IP assays in cells with three different statuses of DNAJA2 (WT, knockout, and over-expression) using either an IRβ or AP2B1 antibody. The results consistently revealed that the intensity of the AP2B1-IRβ interaction is inversely correlated to the expression level of DNAJA2 (Fig. 4F, G). Similar results were observed when the PLA assay was employed to quantify the AP2B1-IRβ interaction in WT, DJ2$^{-/-}$ and DJ2$^{OE}$ 4T1 cells (Fig. 4H, I). Collectively, these data confirm that DNAJA2 counteracts the AP2-IR interaction to prevent spontaneous IR endocytosis.

To confirm if DNAJA2 regulates insulin signaling by counteracting AP2-mediated IR endocytosis, we measured IRβ levels on the PM and insulin stimulated pIRβ levels in WT, DJ2$^{-/-}$ and *DNAJA2/AP2B1* double knockout (DKO) 4T1 cells. As shown in Fig. 4J, K and Supplementary Fig. 5F, G, AP2B1-deficiency restores both the PM localization of IR and insulin signaling in DJ2$^{-/-}$ cells, indicating that DNAJA2 deficiency impairs insulin signaling via AP2-mediated IR endocytosis.

### The J-domain of DNAJA2 is required for blocking the AP2-IR interaction

Next, we aimed to investigate the interaction between AP2 and the IRβ subunit, with a particular focus on the regulatory role of DNAJA2. We identified four putative AP2 binding motifs within the intracellular compartment of IRβ (Supplementary Fig. 6A), with three being located in the tyrosine kinase domain (TKD) and one in the C-terminal domain (CTD)[23]. Our data showed that mutating the first motif in the TKD domain (LL1025-1026AA) dramatically blocks IR-AP2B1 interaction and inhibits IR endocytosis (Fig. 5A–C) without affecting pro-IR processing and maturation (Supplementary Fig. 6B, C). This result drove our hypothesis that DNAJA2 shields these motifs, thereby preventing AP2 from binding to IRβ. To test this postulate, we ectopically expressed HA-tagged constructs of IRβ in HeLa cells, including the full-length (FL), CTD-truncated (ΔCD) IRβ, and IRβ truncated at both the TKD and CTD (ΔTCD). Co-IP assays were then performed using an HA antibody to determine which domain of IRβ interact with DNAJA2. As depicted in Fig. 5D, the interaction between IRβ and DNAJA2 remains intact and is even enhanced upon deletion of the CTD. However, the interaction is significantly attenuated when both the CTD and TKD are truncated. This suggests that DNAJA2 binds to the TKD of IRβ, thereby shielding the majority of critical AP2 binding motifs located within IRβ.

Previous studies demonstrated that the client binding domain (CBD) of DNAJA2 (Supplementary Fig. 6D) is essential for its interaction with client proteins[24]. To investigate whether other domains of DNAJA2 are involved in its interaction with IRβ, we expressed full-length (FL) DNAJA2, as well as constructs with deletions in the J domain (ΔJ) or C-terminal domain (ΔC) (Fig. S6D), in DJ2$^{-/-}$ HeLa cells. Co-IP experiments were then conducted using an IRβ antibody to assess these interactions. The results, shown in Supplementary Fig. 6E, indicate that deletion of the J domain significantly attenuates the interaction between IRβ and DNAJA2, whereas the deletion of the C-terminal domain does not affect this interaction. These findings were corroborated by PLA assays, which produced similar results (Fig. 5E, F). Collectively, these data suggest that the J domain of DNAJA2 is crucial for its interaction with IRβ.

Structure simulation and docking analysis revealed that both the CBD and J domains interact with the TKD domain of IRβ, synergistically locking IRβ (Supplementary Fig. 6F). To further investigate the functional requirement of the DNAJA2 J domain in counteracting the AP2-IR interaction, we performed co-IP assays in DJ2$^{-/-}$ HeLa cells expressing DNAJA2 FL, ΔJ, or ΔC using either IRβ or AP2B1 antibody. As shown in Fig. 5G and Supplementary Fig. 6G, the interaction between IRβ and

AP2B1 was significantly inhibited in the presence of FL DNAJA2 or DNAJA2 with the C-terminal domain truncated, but not in the presence of J domain-deleted DNAJA2. This indicates that the J domain is essential for counteracting the AP2-IRβ interaction.

To determine if the J domain is required for preventing IR endocytosis, we performed immunofluorescence analyses to examine the subcellular localization of IRβ in WT, DJ2$^{-/-}$, and DJ2$^{-/-}$ HeLa cells rescued with FL or ΔJ DNAJA2. The IRβ signal was predominantly located on the PM in WT and DJ2$^{-/-}$ HeLa cells rescued with FL DNAJA2. In contrast, the majority of IRβ signal was found in the intracellular compartment in DJ2$^{-/-}$ and DJ2$^{-/-}$ HeLa cells rescued with ΔJ DNAJA2 (Fig. 5H, I). Collectively, these data demonstrate that the J domain of DNAJA2 is crucial for counteracting the AP2-IR interaction and preventing spontaneous IR endocytosis.

### DNAJA2 dysregulation is associated with glucose metabolic disorders

As we observed broad effects of DNAJA2 on IR endocytosis and insulin signaling in various cell line derived from different tissues (Figs. 2–4 and Supplementary Figs. 3, 4), we hypothesized that DNAJA2 may have broad effects on metabolism in different tissues or organs. To evaluate the role of DNAJA2 in glucose metabolic disorders, we first analyzed *DNAJA2* expression levels from a published dataset (GSE25724[25]) in human islets isolated from T2DM patients and non-diabetic individuals. As shown in Fig. 6A, *DNAJA2* expression was downregulated in islets from T2DM patients compared to non-diabetic individuals. Additionally, analysis of the correlation between *DNAJA2* expression and body mass index (BMI) from another published dataset (GSE20966[26]) revealed that *DNAJA2* expression in beta cells was inversely correlated with BMI, particularly in T2DM patients (Fig. 6B). We then examined *DNAJA2* expression patterns in a dataset (GSE26168[27]) derived from human samples. The data indicated that *DNAJA2* expression was significantly downregulated in the blood of T2DM patients than that of non-T2DM controls (Fig. 6C). Consistent with these findings, analysis of the mouse diabetes database (http://diabetes.wisc.edu/) showed that *DNAJA2* expression in islets was significantly lower in obese T2DM mice compared to their lean counterparts (Fig. 6D), aligning with the observations in humans (Fig. 6B). Conversely, *DNAJA2* was upregulated in the adipocytes of obese mice compared to lean controls (Fig. 6E) Collectively, these data indicate that dysregulation of DNAJA2 is associated with glucose metabolic disorders, such as T2DM. The downregulation of DNAJA2 in islets and its inverse correlation with BMI in T2DM patients, along with its upregulation in adipocytes and hepatocytes in response to obesity and HSD, highlight a complex role of DNAJA2 in glucose metabolism and T2DM pathophysiology.

## Discussion

The complexity of human metabolic disorders, including T2DM, poses significant challenges for pre-intervention strategies and patient outcomes. Although it has been established for decades that insulin resistance is the primary reason for T2DM onset and progression, the factors and underlying mechanisms contributing to insulin resistance remain complex and are not fully understood[14,28,29]. In this study, we demonstrate that DNAJA2 dysregulation is associated with metabolic phenotypes, including T2DM and obesity, in humans and mice (Fig. 6). Mechanistic investigations revealed that DNAJA2 deficiency impairs insulin signaling and induces insulin resistance in mouse models. Therefore, our findings provide mechanistic insights into insulin resistance and glucose metabolism, establishing DNAJA2 as a biomarker for T2DM predisposition and pathogenesis.

HSPs, as the major chaperones and stress regulators in human cells, play critical roles in insulin resistance and progression of T2DM[4,5,15,16]. Among these, HSP70 has been extensively studied and is known to participate in multiple aspects of T2DM onset and

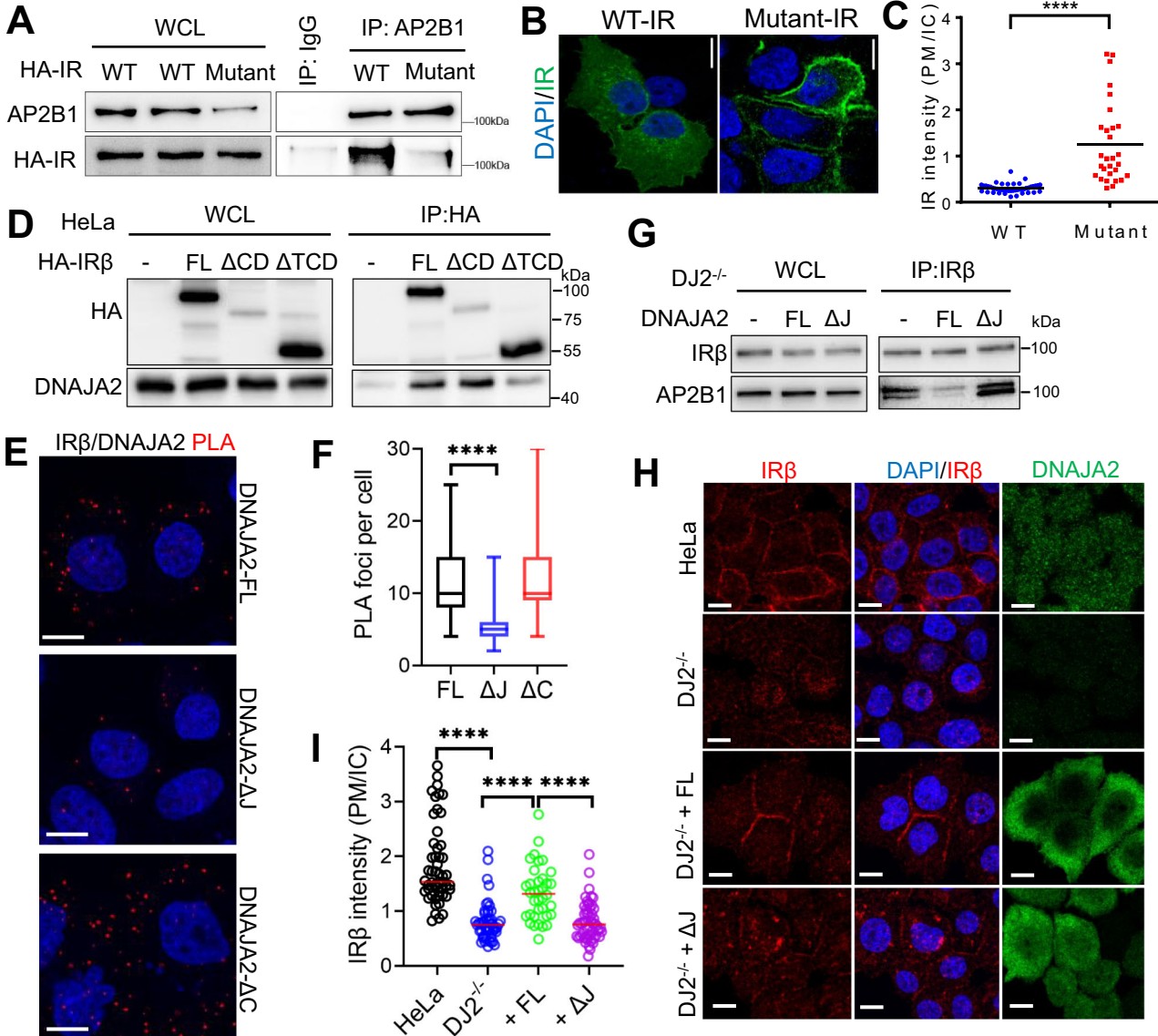

**Fig. 5 | The J domain of DNAJA2 inhibits spontaneous IR endocytosis by blocking the AP2-IR interaction. A** Co-IP assay showing the interactions between AP2B1 and WT or mutant HA-tagged IR. WT or mutant IR (AP2 binding motif 1 mutant) was expressed in HeLa cells and an AP2B1 antibody was used for IP experiments. **B** Immunoflorescence analyses of the subcellular localization of WT and mutant IR. **C** Quantifications of the ratios of IR localized on the plasma membrane (PM) to those in intracellular compartment (IC), as shown in (**B**). **D** Co-IP assay showing the interactions between HA-tagged IRβ fragments and DNAJA2 in HeLa cells expressing full-length (FL) IRβ or an IRβ deleted of the C-terminal domain (ΔCD) or both the C-terminal and tyrosine kinase domains (ΔTCD). Representative images (**E**) and quantifications (**F**) of the PLA assay showing the interaction between IRβ and various forms of DNAJA2s (FL DNAJA2, or J domain-deleted (ΔJ) DNAJA2 or C terminal domain-deleted (ΔC) DNAJA2. **G** Co-IP assay showing the interaction between IRβ and AP2B1 in DJ2$^{-/-}$ HeLa cells expressing empty vector (-), FL or ΔJ DNAJA2. **H** Immunoflorescence analysis of the subcellular localization of IRβ in WT and DJ2$^{-/-}$ HeLa cells, and DJ2$^{-/-}$ HeLa cells expressing FL or ΔJ DNAJA2. **I** Quantifications of the ratios of IR localized on the plasma membrane (PM) to IR in intracellular compartment (IC), as shown in (**H**). Scale bar, 10 μm. Data are shown as means ± SEM (**C**) or median (**F**). P values were determined by two-tailed unpaired t test with Welch's correction. ****$p$ < 0.0001. Source data are provided as a Source Data file.

development[15]. Interestingly, extracellular and intracellular HSP70 exhibit different functions in modulating insulin signaling, making HSP70 a potential therapeutic target for T2DM[4,15]. Recently, emerging clinical evidence has indicated that members of the HSP40 family are widely-associated with T2DM phenotypes[16], although the functional mechanisms of HSP40 protein in metabolism and diabetes remain unclear. DNAJB3, for instance, has been reported to affect the stress-responsive kinases JNK1 and IKKβ to modulate insulin resistance in vitro[30,31]. In this study, we reveal that DNAJA2, a member of DNAJA subfamily, regulates insulin signaling and glucose homeostasis by inhibiting spontaneous IR endocytosis under basal conditions in

mouse models. DNAJA2 binds to the intracellular TKD domain of IRβ and shields the putative AP2 binding motifs, thereby blocking the AP2-IR interaction and AP2-mediated IR endocytosis (Fig. 7A). In the presence of DNAJA2, IR is properly localized on the PM and can be efficiently activated by extracellular insulin to transduce signaling cascade and maintain metabolic homeostasis (Fig. 7A). Conversely, the absence of DNAJA2 significantly reduces the presence of IR on the PM, leads to insulin resistance and disrupts homeostatic metabolism and growth (Fig. 7B). Our study establishes a model for understanding how HSP40 proteins execute fundamental roles in regulating IR endocytosis, insulin signaling transduction, and glucose metabolism. This model

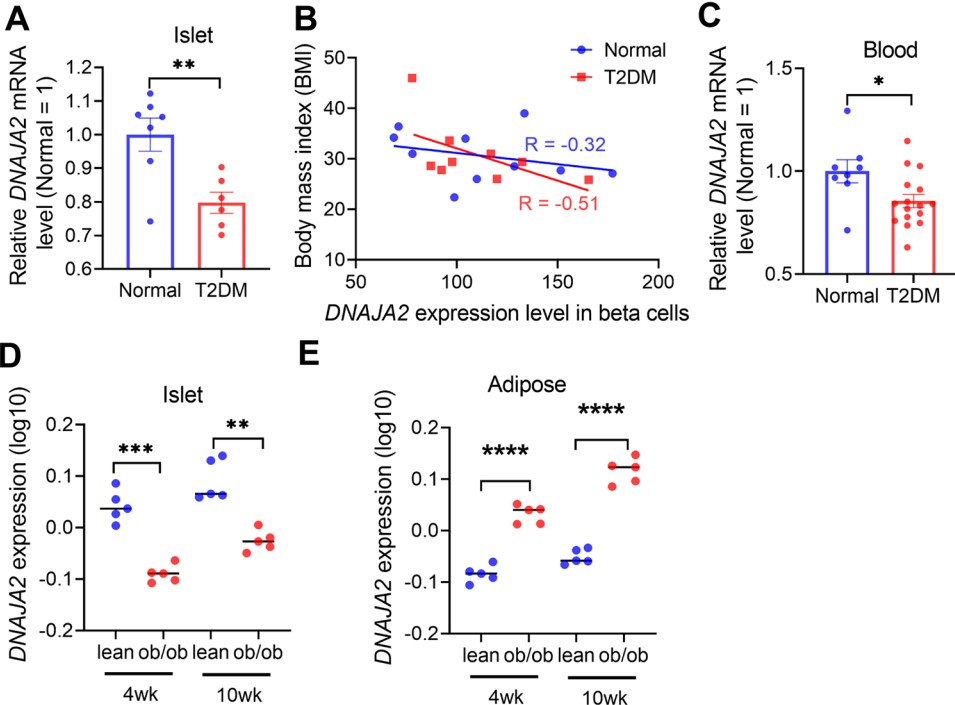

**Fig. 6 | DNAJA2 dysregulation is associated with glucose metabolic disorders. A** Relative mRNA levels of *DNAJA2* in islets from normal and type 2 diabetes mellitus (T2DM) individuals. **B** Correlations between body mass index (BMI) and *DNAJA2* mRNA levels of beta cells in normal individuals and T2DM patients. **C** Relative mRNA levels of *DNAJA2* in blood samples from normal and T2DM individuals. *DNAJA2* mRNA levels of islets (**D**) and adiposes (**E**) in obese T2DM mice and their lean counterparts. Data are shown as means ± SEM (**A**, **C**) or median (**D**, **E**). P values were determined by two-tailed unpaired t test with Welch's correction. *$p < 0.05$; **$p < 0.01$; ***$p < 0.001$, ****$p < 0.0001$. Source data are provided as a Source Data file.

highlights the critical function of DNAJA2 in maintaining insulin receptor localization and signaling, offering insights into the molecular mechanisms underlying insulin resistance and T2DM pathogenesis.

HSP40 proteins, particularly those in the DNAJA and DNAJB subfamilies, possess relatively conserved CBD domains and similar J domains, suggesting potential similarities in their roles in controlling IR endocytosis, insulin signaling, and glucose metabolism. However, genetic knockout mouse models of various HSP40 members exhibit different phenotypes[10], indicating distinct molecular functions. This is consistent with our observation that the J domain of DNAJA2 is essential for inhibiting IR endocytosis, as the J domain is diverse among HSP40 members and determines their specificity[32,33]. Unlike DNAJA2, DNAJB3 regulates insulin signaling and insulin resistance by modulating the stress-responsive kinases JNK1 and IKKβ[30,31]. These observations suggest that the structural diversity and subcellular localization of different HSP40 members determine their specific roles in insulin resistance and T2DM. The differences in their molecular functions underscore the complexity of HSP40 proteins in regulating metabolic processes and highlight the need for further research to elucidate their distinct contributions to glucose metabolism and diabetes pathogenesis.

Comparing to whole-body deletion of DNAJA2, which results in neonatal lethality and mild growth retardation (Fig. 1), the liver-specific DNAJA2 ablation does not induce these defects. This suggests that DNAJA2 also plays critical roles in other tissues or organs, in addition to its function in insulin signaling and glucose metabolism in the liver. For example, DNAJA2 regulates lysosomal pH and lung respiration in mouse models[11], which may contribute to the neonatal lethality. Additionally, DNAJA2 promotes genome stability[7,8], and deficiency in this role can lead to chronic inflammation, cellular senescence, and premature aging[34,35]. We also observed a significant reduction in fat mass in whole body $DJ2^{-/-}$ mice compared to WT, suggesting that

DNAJA2 may regulate IR and other receptors in adipocytes, thereby affecting fat tissue growth. Future studies are needed to investigate the tissue-specific functions of DNAJA2 in tissue development and homeostasis.

Homeostatic insulin signaling requires the proper localization of IR on the PM, where it can be efficiently activated without over-activation. This balance is maintained through IR internalization and recycling to the PM. Therefore, cells have evolved an accurate system to control IR internalization via endocytosis. Our findings show that DNAJA2 acts as an inhibitor of IR endocytosis by binding to the TKD domain of IRβ in the basal state. In contrast, the spindle checkpoint modules MAD2, BUBR1, and Inceptor (insulin inhibitory receptor) and EphB4 (Eph receptor B4) facilitate IR endocytosis[18,20,36]. Specifically, MAD2 directly binds to the C-terminal domain of IRβ and promotes AP2 recruitment through the BUBR1-AP2 interaction. Our data indicate that the C-terminal domain of IRβ inhibits the DNAJA2-IR interaction (Fig. 5A), suggesting potential competition between the MAD2-BUBR1 module and DNAJA2 in controlling IR endocytosis. It is worth investigating the interplay between DNAJA2 and the spindle checkpoint regulators in controlling IR internalization and insulin signaling in future studies.

Notably, DNAJA2 has recently been shown to play critical roles in regulating DNA repair and mitotic division, and defects in DNAJA2 lead to genome instability and aberrant chromosome segregation[7,8], abnormalities that are well documented to induce tumorigenesis. However, this seems to be in conflict to the data shown in this study, where *DNAJA2* knockout mice exhibited neonatal lethality, but had no cancer development. We believe that this discrepancy can be explained by the diverse functions of DNAJA2 and required duration for developing a given disease. During embryogenesis, DNAJA2 is essential for regulating insulin signaling to provide necessary nutrients for organ/tissue growth, which only takes 21 days. However, developing a cancer, e.g., due to loss of a DNA repair pathway, takes a

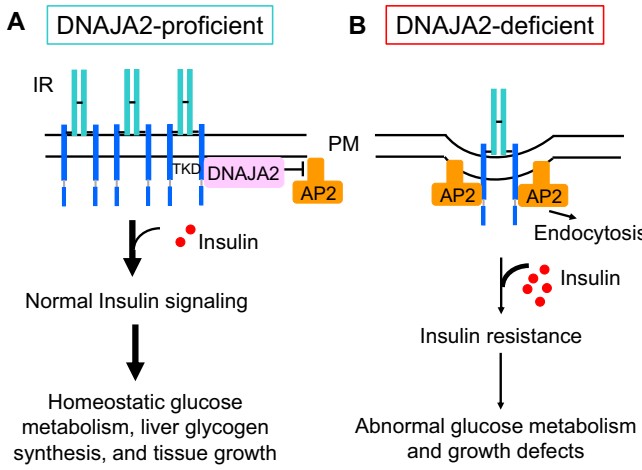

**Fig. 7 | A working model describing the mechanism by which DNAJA2 regulates insulin signaling and glucose metabolism. A** In DNAJA2-proficient hepatocytes, DNAJA2 directly binds to the intracellualr TKD domian of IRβ, which blocks the IRβ-AP2 interaction and inhibits AP2-mediated IR endocytosis. Consequently, the majority of IR are localized on the plasma membrane (PM), and can be readily activated by the coming insulin efficiently. The normal insulin signaling then drives homeostatic glucose metabolism, glycogen synthesis and tissue growth. **B** In DNAJA2-deficient cells, the AP2 complex binds to the membrane IR and promotes IR interlization through clathrin-mediated endocytosis, thereby significantly reducing the IR localization on the PM. As a result, the insulin is unable to activate the insulin signaling cascade, which leads to insulin resistance, and disrupts glucose metabolism including glycogen synthesis, and tissue growth.

minimum of 4–6 months[37]. Thus, it is highly possible that mice with conditional *DNAJA2* knockout can develop tumors if sufficient time is provided. This possibility awaits future investigations.

## Methods

### Antibodies, chemicals and other reagents
All the antibodies, chemicals, oligonucleotides and plasmids used in this study are listed in Supplementary Table 1.

### Cell lines and cell culture
Human cell lines HeLa, HepG2-IR (kindly provided by Dr. Eunhee Choi), and mouse breast cancer cell line 4T1 as well as colon adenocarcinoma cell MC38 were used in this study. Unless stated otherwise, all cells were cultured in a 37 °C incubator supplemented with 5% $CO_2$. HeLa cells were grown in RPMI 1640 media supplemented with 10% FBS. HepG2-IR, 4T1, and MC38 cells were cultured in Dulbecco's Modified Eagle Medium (DMEM) supplemented with 10% FBS. All knockout cell lines were generated using CRISPR-Cas9 technologies. Cells expressing DNAJA2 and IR fragments were constructed by transfecting the recombinant plasmids containing the corresponding coding sequences, followed by puromycin selection, single colony isolation, and verification.

### Mouse strains
The whole-body *DNAJA2* knockout heterozygotes (C57BL/6NCrl-Dnaja2em1(IMPC)Mbp/Mmucd) were purchased from Mutant Mouse Resource and Research Center (MMRRC). The Alb-Cre mice (B6.Cg-Speer6-ps1<Tg(Alb-cre)21Mgn>/J) were purchased from the Jackson Laboratory. The *DNAJA2*$^{loxp/+}$ mice were generated by targeting mouse ES cells and blastocyst injection in SHANGHAI MODEL ORGANISMS. The *DNAJA2*$^{loxp/+}$ mice were backcrossed onto C57BL/6 background for three generations. The liver-specific *DNAJA2* knockout mice (CKO) were generated by crossing *DNAJA2*$^{loxp/+}$ mice with Alb-Cre mice. All the mice were maintained in a specific-pathogen-free animal facility, and all

experiments were conducted according to regulations of the Institutional Animal Care and Use Committee of the Chinese Institutes for Medical Research, Beijing.

### Tissue histology and immunohistochemistry
Mouse liver tissues were fixed in 1% periodic acid in 10% NBF for hematoxylin and eosin (H&E) staining and periodic acid–Schiff (PAS) staining performed by the Molecular Pathology Core at Chinese Institutes for Medical Research, Beijing. Images were captured with a Keyence BZ-X700 fluorescent microscope.

The immunohistochemistry (IHC) analysis of IR in liver sections was performed as described[38]. Briefly, sections were first dewaxed, subjected to antigen retrieval with 10 mM sodium citrate (pH 6.0), permeabilized with gelatin (0.2% w/v)/Triton(0.25% v/v) in PBS and blocked with 5% BSA. Then the blocked sections were incubated with primary antibodies anti-IRβ (Millipore, MABS65) and anti-RAB7 (Cell Signaling, #9367) antibodies overnight, followed by fluor-conjugated secondary antibodies for 2 h at room temperature. After the final wash steps with 10 mM CuSO4/50 mM NH4Cl solution and distilled water, slides were mounted with DAPI solution. Images were captured using a Leica TCS SP8 confocal microscope.

### Measurements of liver glycogen, serum insulin and blood glucose
Mouse liver tissues were homogenized in distilled water on ice and then boiled for 5 min to precipitate insoluble debris. The soluble supernatants were used for glycogen measurement with the glycogen assay kit (Millipore Sigma) according to the manufacturer's instructions.

To quantify serum insulin level, mice were sacrificed and their blood were collected from inferior vena cava. The whole blood was allowed to form clots at room temperature for at least 30 min, then centrifuged at 2000 × $g$, 4 °C for 20 min to collect the serum. Serum insulin level was measured using the Ultra Sensitive Mouse Insulin ELISA Kits (Crystal Chem, Inc.) according to the manufacturer's instructions.

Blood glucose levels from tail bleeding were measured with a glucometer (Clarity BG1000).

### Glucose rescue experiment
The glucose rescue experiment was performed as described[20]. *DJ2*$^{-/-}$ newborns and their littermates were administered with 50 µl 10% D-glucose subcutaneously after birth and every 6 h thereafter up to 24 h. The survival of all the newborns was monitored accordingly.

### Glucose and insulin tolerance tests
The glucose tolerance test in newborns was performed as described[20]. *DJ2*$^{-/-}$ newborns and their littermates were injected with 50 µl 10% D-glucose subcutaneously and then were sacrificed at a predefined time point (0, 30, 60 or 120 min) to measure blood glucose levels using a glucometer.

The glucose tolerance and insulin tolerance tests in adult mice were conducted as described[18]. For glucose tolerance test, mice were fasted for 15 h before their blood glucose levels (T = 0) were measured using a glucometer. All the mice were then injected with 2 g of glucose per kg of body weight intraperitoneally, and their blood glucose levels were measured at 15, 30, 60, and 120 min after glucose injection. For insulin tolerance test, mice were fasted for 2 h before they were injected with human insulin intraperitoneally at 1 U/kg body weight. Then their blood glucose levels were measured at each time point (0, 15, 30 and 60 min).

### Glycogen synthase (GS) activity assay
Mice were fasted for 2 h before they were injected with human insulin intraperitoneally at 1 U/kg body weight. 10 min later, livers were

dissected and 100 mg liver tissues were homogenized in 1 ml tissue lysis buffer on ice. Then the tissue lysate was centrifuged at $10000 \times g$, 4 °C for 10 min to collect the supernatant. The clear lysate was subjected to GS activity measurement using the Glycogen Synthase Activity Assay Kit (Boxbio) according to the manufacturer's instructions.

## Immunoblot analyses of insulin signaling

To detect the insulin signaling in newborns, the newborn pups were starved for 4 h before their livers were harvested for western blot analysis. For in vivo insulin signaling, adult mice were fasted for 4 h before they were injected with insulin intraperitoneally at 1 U/kg body weight. The mice were sacrificed 10 min after insulin injection, and their livers were removed and processed for western blot analysis. For in vitro insulin signaling, cells were serum-starved overnight before being incubated with an insulin solution at the indicated concentrations or times.

## Isolation of primary hepatocytes

Primary hepatocytes were isolated from WT and $DJ2^{-/-}$ adult mice (10 weeks) using an improved two-step collagenase perfusion protocol as described[39]. The isolated hepatocytes were firstly plated on collagen-coated cell culture plates with plating media (DMEM low glucose, 5% FBS and 1% penicillin-streptomycin solution). After 3–5 h incubation, the media was changed to maintenance media (Williams E media, 2 mM glutamine and 1% penicillin-streptomycin solution) for 24 h before cells were used for experiments.

## Indirect immunofluorescence

Cells cultured on the cover slides were fixed with 4% paraformaldehyde in PBS for 10 min at room temperature, and permeabilized in 0.25% Triton X-100 for 10 min, followed by blocking with 5% BSA in PBS for 30 min. The fixed cells were then incubated with primary antibodies and secondary antibodies each for 2 h at room temperature. The slides were mounted with DAPI solution after final washing with PBS. All images were captured using a Leica TCS SP8 confocal microscope, and analyzed and quantified using the NIH ImageJ software. The whole cell intensity (WC) and intracellular intensity (IC) of IR were measured using ImageJ. Subsequently, the plasma membrane intensity (PM) of IR was calculated by subtracting IC from WC.

## Membrane and cytoplasmic protein extraction

Membrane protein isolation was performed using the Membrane and Cytoplasmic Protein Extraction kit (Sangon Biotech) according to the manufacture's instructions. Briefly, $5 \times 10^6$ cells were washed with cold PBS twice and homogenized in 1 ml buffer A on ice until more than 90% cells were broken (checked by microscope). The cell lysate was centrifuged at 4 °C, $1000 \times g$ for 10 min to retain the supernatant. Then the supernatant was centrifuged at 4 °C, $22000 \times g$ for 1 h. The supernatant was kept as cytoplasmic proteins while the pellet was resuspend in 500 µl buffer B on ice for 30 min with 5–6 times vortex. Finally, the lysate was centrifuged at 4 °C, $20000 \times g$ for 10 min to collect the supernatant which contains membrane proteins.

## Endocytosis assay

Insulin endocytosis assay was performed as described[18]. WT and *DNAJA2* knockdown HepG2 Cells cultured on slides were starved overnight and stimulated with 50 nM Cy3-labeled insulin (NANOCS) for 0, 10, 20, and 30 min. As a control, another batch of cells were similarly treated with 25 µg/ml Alexa-568-labeled transferrin (Invitrogen). Cells were then washed three times with PBS and fixed with 4% paraformaldehyde on ice, followed by permeabilization in 0.25% Triton X-100 for 10 min. After briefly washing with PBS, cells were mounted with DAPI solution (Invitrogen). Images were taken using a

Leica TCS SP8 confocal microscope, and analyzed using ImageJ. The adjusted intensity at each time point (10, 20 and 30 min) was calculated by subtracting the intensity of time zero from that of the specific time.

## Transfection and chemical treatment

Recombinant plasmids were transfected into cells using jetPRIME® transfection reagent (PolyPlus) for 36–48 h. SiRNA transfection was performed using Lipofectamine™ RNAiMAX transfection reagent (Invitrogen) for 48–72 h. Cells were treated with 50 µM dynasore (MedChemExpress) for 4 h before performing immunofluorescence or co-immunoprecipitation assays.

## Co-immunoprecipitation and Western blots

For cultured cells, cell pellets were incubated with lysis buffer (50 mM Tris-HCl pH 7.4, 150 mM NaCl, 1% (v/v) NP-40, 1 mM EDTA, 1% sodium deoxycholate) supplemented with protease inhibitor cocktail on ice for 30 min, followed by centrifuge at $15,000 \times g$, 4 °C for 15 min. For mouse liver tissues, 100 mg of tissues were homogenized and incubated in 1 ml lysis buffer supplemented with protease inhibitor cocktail and 100 µM cytochalasin B for 60 min. The lysates were then centrifuged and filtered to collect clear supernatants. The clear cell lysates or tissue supernatants were incubated with a primary antibody overnight, and the protein-antibody conjugates were incubated with Pierce™ Protein G Agarose beads (Thermo Scientific™) for 1–2 h. After three times of washing with lysis buffer containing increased concentrations of NaCl, the beads were resuspended and boiled with loading buffer, and the samples were subjected to SDS-PAGE and Western blot analysis.

## PLA assay

Cells grown on the cover slides were fixed with 4% paraformaldehyde in PBS for 15 min at room temperature and permeabilized in 0.25% Triton X-100 for 10 min. The fixed cells were then blocked with 5% BSA for 30 min and incubated with primary antibodies for 2 h at room temperature. Then the proximity ligation assays were performed using Duolink In Situ Red Starter kit (Sigma-Aldrich) according to the manufacturer's instructions. Images were captured using a Leica TCS SP8 confocal microscope. Quantification was conducted using ImageJ.

## Protein docking analysis

The structure of tyrosine kinase domain (TKD) in IR and the alpha-fold predicted DNAJA2 structure were downloaded from the PDB database. The protein simulation and docking analysis was performed using the online MDockPP[40] server (https://zougrouptoolkit.missouri.edu/MDockPP/index.php) with default parameters. The resulted top 1 complex structure was viewed and displayed using VMD software.

## Statistical analysis

Statistical analyses were performed in GraphPad Prism 9.0 using unpaired Student's *t* tests, except for Fig. 6F using Wilcoxon Rank Sum test. Data were shown as means ± SEM unless specified in the figure legends. A value of $p < 0.05$ was considered statistically significant (ns, no significance, *$p < 0.05$, **$p < 0.01$, ***$p < 0.001$, ****$p < 0.0001$).

## Reporting summary

Further information on research design is available in the Nature Portfolio Reporting Summary linked to this article.

## Data availability

All data are available in the main text or the supplementary materials. Source data are provided with this paper. *DNAJA2* expression data are downloaded from published datasets (GSE2572425[25], GSE2096626[26] and GSE26168[27]). *DNAJA2* expression data in mice are downloaded

from the Mouse Diabetes Database (http://diabetes.wisc.edu/). Source data are provided with this paper.

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

## Acknowledgements

We thank Dr. Guo-Min Li (Chinese Institutes for Medical Research, Beijing) for stimulatory discussion and suggestions and Dr. Eunhee Choi (Columbia University) for reagents. This work was supported by the National Natural Science Foundation Youth Program (8250033613) and Chinese Institutes for Medical Research funds to Y.H.

## Author contributions

Conceptualization: Y.H., Funding acquisition and supervision: Y.H., Experimental performance and analysis: Y.Q., W.W., and K.L., Writing: Y.H., W.W. and A.J.D.

## Competing interests

The authors declare no competing interests.
