## [Transparent Peer Review file · Nature Communications]

Heat shock protein DNAJA2 controls insulin signaling and glucose homeostasis by preventing spontaneous insulin receptor endocytosis

Corresponding Author: Dr Yaping Huang

Version 0:

Reviewer comments:

Reviewer #1

(Remarks to the Author)

The authors analyze the effect of DNAJA2, a kind of heat shock protein, on neonatal lethality and glucose homeostasis. They describe a role of DNAJA2 deficiency in facilitating clathrin-mediated endocytosis of InsR. The authors first show that DNAJA2 binds to insulin receptor (InsR). DNAJA2-specific liver knock-out decreased insulin sensitivity and glucose tolerance in chow-fed normal mice. Overall, this is an interesting and important study.

Points need to be addressed:

1. Given that DNAJA2 deficiency (whole body ko and liver-specific ko) had a strong effect on the size of newborn pups (Fig.S1A, and C-D), indicating that these mice have some problems with development and growth. So, the phenotype observed in liver-specific ko mice might be the secondary effect of growth defects. Use the inducible liver-specific ko mice or adenovirus to delete hepatic DNAJA2 when the mice grow up. This is very important.
2. Conclusion of the first paragraph in the discussion section: "Therefore, our findings provide new mechanistic insights into insulin resistance and glucose metabolism, establishing DNAJA2 as a new biomarker for T2DM predisposition and pathogenesis." Right now, the phenotype is observed on chow-fed normal mice. To support this conclusion, they need to evaluate the phenotype of liver DNAJA2 ko on HFD feeding. plus these mice are better to be inducible ko. Additionally, they need to prove that hepatocyte DNAJA2 is downregulated in the insulin-resistant and/or diabetic models (high fat diet fed mice or ob/ob mice or db/db mice) compared to controls.
3. The protein levels of DNAJA2 in the liver and other tissues.
4. The authors analyzed the DNAJA2 expression levels from published datasets in the human samples with/without diabetes. DNAJA2 expression was downregulated in the islets from T2DM patients compared to non-diabetic individuals. However, DNAJA2 was upregulated in the adipocytes of obese mice compared to lean controls (Figure 6E), and a high-sugar-and-high-fat diet (HSD) also induced higher DNAJA2 expression in mouse hepatocytes (GSE18236528) (Figure 6F). The result of the upregulation of DNAJA2 in the hepatocytes contradicts the finding that DNAJA2 deficiency led to insulin resistance (Figure 1). Probably, the protein levels and mRNA levels are not consistent in the case of DNAJA2. It's better to measure the DNAJA2 protein levels in the liver samples of human subjects with diabetes or not.
5. The first sentence (Line 68-78) of one paragraph : "The observation that DNAJA2 deficiency induces genomic instability prompted us to investigate its role in spontaneous tumorigenesis in mouse models." This paragraph just showed the results of the growth and survival. There is no data related to tumorigenesis. Please revise the first sentence of this paragraph.
6. Conclusion of the third paragraph in the results section: "Therefore, our data indicate that insufficient glycogen storage in livers is a major cause of DJ2-/neonatal lethality." I think this conclusion is not consistent with the finding in DNAJA2-specific liver knock-out mice, which are born at the normal Mendelian ratio and survived to adulthood. In the Discussion Section, the authors then listed another possibility (Line 323-326: "in addition to its function in insulin signaling and glucose metabolism in the liver. For example, DNAJA2 regulates lysosomal pH and lung respiration in mouse models¹¹, which may contribute to the neonatal lethality.") the authors need to revise the conclusion.

7. The data about DNAJA2 and InsR interaction is not convincing. (1) There is even a band in the negative control group in the Co-IP experiments (Fig.S5A, S5B). (2) The signal in the PLA experiments is not strong. (3) The co-IP experiments were done in an overexpression system? If so, please repeat the Co-IP experiments to evaluate the endogenous binding in the primary hepatocytes.
8. Does DNAJA2 affect InsR synthesis or turnover? How does pro-InsR look like?
9. The data showed that DNAJA2 deficiency promoted InsR endocytosis and location in late lysosome complex (Rab 7 staining results). Generally, in the late lysosome complex, the protein will degrade. As one literature (<https://doi.org/10.1038/s42255-022-00634-5>) showed EphB4 stimulated InsR endocytosis and InsR/Rab7 colocalization in the late lysosome, which resulted in InsR degradation. But the data in Fig.2E and 2F indicate that DNAJA2 deficiency did not affect InsR degradation. Why is that?
10. I am not quite convinced about the InsR endocytosis upon DNAJA2 deficiency and the authors should show this with an alternative method. i.e. surface biotinylation, that InsR levels at the cell membrane are changed upon DNAJA2 LOF and GOF.
11. Measure the InsR protein level on the cell membrane.
12. It looks to me that DNAJA2 deficiency has a strong effect on the InsR phosphorylation (fig.2A, 2C, 2E, 2F). It's most likely that DNAJA2 deficiency has a direct effect on InsR phosphorylation besides InsR endocytosis.
13. The impaired InsR phosphorylation due to DNAJA2 deficiency will lead to insulin resistance in the hepatocytes. Function assay like gluconeogenesis should be performed in primary hepatocytes.
14. It's nice that the authors discussed other proteins (MAD2, Inceptor), which also facilitate InsR endocytosis. Please include EphB4, which was shown to mediate the effect of insulin on InsR endocytosis.
15. Please indicate age, sex, number of mice in the figure legend. Also, for cell experiments, provide the treatment detail (time, concentration) in the figure legend.
16. For the WB results, annotate them with molecular weight labels, so the readers could have a general idea about the molecular weight of detected proteins.
17. Glucose and insulin tolerance tests in Fig 1G and H are consistent with the overall hypothesis of the DNAJA2-InsR interactions. Since these significant effects are small, the authors attempt to improve the impact of their figures by starting the y-axis some distance from zero. I think it is generally best to start the y-axis at zero, so comparisons are easier for the reader throughout the manuscript.

Reviewer #2

(Remarks to the Author)

This study examines the role of the DNAJA family of chaperones, DNAJA2 (DJ2), in regulation of whole-body insulin signaling and glucose homeostasis, as well as in regulation of insulin action and signaling focusing on the liver. For in vivo studies, two mouse models were examined, a whole-body DJ2 $-/-$ model as well as a liver-specific DJ2 knockout model (CKO). Whole-body DJ2 $-/-$ exhibited neonatal lethality with largely normal organ development, with the exception of liver abnormalities that included reduced glycogen storage. The neonatal viability phenotype was rescued by subcutaneous injection of glucose, indicating that disruptions of glucose metabolism contributed to the lethality phenotype in DJ2 $-/-$ animals. CKO mice exhibited normal liver development and did not exhibit neonatal loss of viability, but also demonstrated disruptions in glucose metabolism which included elevated blood glucose and insulin levels in the fed state.

IR signaling (focusing on IR beta phosphorylation) was found to be reduced in both DJ2 $-/-$ liver lysates as well as in liver samples of CKO mice 10 min following injection of a pulse of insulin. Reduced pIR β phosphorylation and reduced levels of phosphorylation of other insulin signaling proteins upon disruption of DJ2 was also observed in hepatocytes isolated from DJ2 $-/-$ animals as well as in HepG2 and other cell lines upon silencing DJ2. In HepG2 cells, silencing DJ2 led to a reduction of IR levels at the plasma membrane (relative to cytosol levels) prior to insulin stimulation (basal state) and also reduced the uptake of fluorescently-conjugated insulin. The decrease in PM/C (plasma membrane to cytosol) levels of IR in DJ2 silenced cells was reversed by treatment with the endocytosis inhibitor dynasore or by silencing the clathrin-mediated endocytosis adaptor protein AP2, suggesting that the loss of DJ2 enhanced clathrin-mediated endocytosis of IR. DJ2 silencing was without effect on the cell surface levels of transferrin receptor or transferrin internalization, suggesting that DJ2 specifically regulated clathrin endocytosis of IR but did not broadly impact regulation of all clathrin endocytosis cargo.

DJ2 co-precipitated with IR β , and silencing or overexpression of DJ2 led to an increase or reduction, respectively, of the association of IR β with AP2, detected by co-IP or PLA. The interaction of IR β with DJ2 detected by co-IP or PLA required the C-terminal and tyrosine kinase domains of IR and the J domain of DJ2. Lastly, some analysis of the expression data of various previous studies seems to suggest that DJ2 exhibits alterations in various tissues that correlates with insulin

resistant or T2DM conditions.

Overall, this is an interesting study that provides new insights into the regulation of insulin receptor signaling, liver function, and whole-body glucose metabolism. The role of DJ2 in the regulation of insulin receptor membrane traffic provides important mechanistic insights to accompany the *in vivo* work using two different DJ2 knockout mouse models that establish the physiological relevance of DJ2 in glucose metabolism and insulin action. The interaction experiments using co-IP and PLA are largely well conducted and provide some insights into how DJ2 may regulate IR endocytosis. Moreover, the experiments that dissect the specific regions of DJ2 that interact with IR are very useful to support a model of more direct action of DJ2 on binding to IR to modulate IR endocytosis, rather than considering more indirect effects of DJ2 on insulin signaling. This study will be of interest to a wide range of cell biology and metabolism researchers, but should first look to address the following comments:

Major comments

- 1) The experiments included in this study probing the interaction of DJ2 with IR β do not allow conclusion of a direct interaction of these two proteins. Both IP and PLA do not allow resolution of direct interactions, which requires study of interaction of recombinant proteins or other methods. With the data included in this study, it appears that IR β and DJ2 interact, but this cannot conclude direct or indirect interactions. Similarly, the model by which AP2 and DJ2 compete for binding to IR depends on direct interactions of each with IR, which has not been demonstrated by the collective experiments included in this study. Either additional experiments should be included to support the model of direct interaction, or the conclusions and interpretation of these results should be tempered.
- 2) Lines 59-61 of the introduction state the following conclusion based on the results of the current study: "The spontaneous endocytosis of IR significantly reduces its plasma membrane localization, causing insulin resistance and disrupting glucose metabolism and homeostasis." The causal link between the increased rate of basal endocytosis of IR and insulin resistance and disrupted glucose metabolism (upon DJ2 perturbation) was not established, and conclusions stating as such should thus be revised. Please review the rest of the manuscript for similar statements about causation that are not supported by the data, including for example lines 301-303.
- 3) Lines 165-176 and Figure 3A-B and S4: Dynasore is not considered a specific inhibitor of clathrin-mediated endocytosis, see PMID: 24046449. The study includes experiments that rely on disruption of clathrin endocytosis with the more selective strategy of silencing AP2, which supports the conclusion that the disrupted membrane traffic of IR is clathrin-endocytosis-dependent, but care should nonetheless be taken when discussing dynasore.
- 4) Are these sites in the CTD and the TKD known if IR to bind AP2? What is the evidence and can any previous studies that established this be cited here (lines 224-225)? Does this mutant CTD-TKD of IR-beta also exhibit reduced internalization, for instance upon stimulation with insulin? This additional experiment would be helpful in establishing support for the model that is presented.
- 5) As an additional extension to the point above, does insulin stimulation trigger a loss of DJ2 interaction with IR? This should be a relatively straightforward experiment to probe (e.g. by co-IP) and would strengthen the conclusion that DJ2 impairs internalization of IR.
- 6) Lines 241-243: What is meant by an endogenous co-IP experiment? This experiment appears to be done in cell that are DJ2 knockouts that have been rescued with various DJ2 constructs, which cannot be readily considered "endogenous". Please provide a clearer explanation of what is meant by "endogenous" here.
- 7) It is not clear what the rationale for examination of the levels of DJ2 expression in various tissues (Figure 6) may be. Many of the studies examined appear to be in islets (pancreatic islets). However, the role of DJ2 in regulation of endocytosis in islets (and indeed in any of the specific cell types of islets) is not clear and not probed by the current study. Moreover, the role of blood levels of DJ2 has not been examined, and so it is very difficult to understand how examining the blood levels of DJ2 in samples obtained from patients with insulin resistance or T2DM reveals information pertinent to this study. This analysis of patient samples could be removed from the manuscript, or alternatively, careful consideration and justification for study of the expression levels of DJ2 in this broad array of tissue samples should be provided.
- 8) Figure 4A, and more importantly Figure 4E show detection of DJ2 with a red arrow. The band indicated is very weak relative to a much stronger band at a higher molecular weight. In Figure 4E, it is expected that DJ2 is absent from the knockout sample, it appears it could be, but the blot shown is not of sufficient clarity and should be replaced.
- 9) Building on comment 1 above about the lack of support for a causal relationship for the role of DJ2 in regulating IR signaling and endocytosis in liver to whole-body glucose metabolism, there is also a lack of support for a causal relationship between DJ2 regulation of IR endocytosis and IR signaling. The latter can be much more readily probed, for instance in cell lines. It would be very useful to examine how either (i) loss of endocytosis by AP2 silencing (ii) loss of endocytosis of IR specifically with a mutant that cannot bind DJ2 but retains binding to AP2 (if this exists) impacts insulin signaling upon silencing of DJ2. If loss of IR endocytosis (e.g. by loss of AP2) can rescue insulin signaling in DJ2-silenced cells, this would significantly support this aspect of the conclusions and model.
- 10) Some consideration of the fate of internalized IR upon DJ2 silencing could be warranted in the discussion. Rab7 should demark late endosomal compartments, such that increased localization of IR to Rab7 compartments upon DJ2 loss of

function might be expected to contribute to IR degradation, but the levels of IR do not appear to change. This point could be addressed in the discussion.

Minor comments:

1) Line 639: domians (misspelling)

2) Line 152: describing the disruption of IR internalization as “unscheduled” endocytosis of IR is unusual – implies that there is a disruption of the timing of IR in response to a specific cue, while this is perhaps better considered as a disruption of the plasma membrane retention of IR in the basal state.

3) Figure 3: the figure caption indicates that panels E-F represent the quantification of internalized insulin and transferrin, but only that of transferrin appears to be shown.

Version 1:

Reviewer comments:

Reviewer #1

(Remarks to the Author)

The authors addressed most of the questions. With regard to the response to question 1 and question 9, it's still not clear.

1. Given that DNAJA2 deficiency (whole body ko and liver-specific ko) had a strong effect on the size of newborn pups (Fig.S1A, and C-D), indicating that these mice have some problems with development and growth. So, the phenotype observed in liver-specific ko mice might be the secondary effect of growth defects. Use the inducible liver-specific ko mice or adenovirus to delete hepatic DNAJA2 when the mice grow up. This is very important.

Thanks for this question. We are sorry for the misleading information. Actually, Fig.S1A, and C-D show only whole-body KO mice, whose growth are affected. Liverspecific KO does not affect the mouse survival, development or growth (Figure S2F). We have the description in the line 115-116: “We observed that CKO mice were born at the normal Mendelian ratio, survived to adulthood (Figure S2F), and were morphologically indistinguishable from WT mice”, which rules out the possibility that the observed phenotype is the secondary effect of growth defects.

Question :Figure S2F shows the survival curves of wild-type (WT), heterozygous (Het), and liver-specific DNAJA2 knockout (CKO) mice. This data does not support the conclusion that the observed phenotype in knockout mice is not a secondary effect of growth defects. Therefore, data similar to that presented in Figures S1C and S1D should be provided.

9. The data showed that DNAJA2 deficiency promoted InsR endocytosis and location in late lysosome complex (Rab 7 staining results). Generally, in the late lysosome complex, the protein will degrade. As one literature (<https://doi.org/10.1038/s42255-022-00634-5>) showed EphB4 stimulated InsR endocytosis and InsR/Rab7 colocalization in the late lysosome, which resulted in InsR degradation. But the data in Fig.2E and 2F indicate that DNAJA2 deficiency did not affect InsR degradation. Why is that?

Thanks for pointing this out. Actually, in most of our WB results, IR levels are lower in DJ2-KO cells or tissues (Figures 2A, 4A, 4B, Figures S3, S4D, S5A), suggesting partial degradation of this protein in KO cells.

Question: Please explain why IR remains unchanged in Figures 2E and 2F.

Reviewer #2

(Remarks to the Author)

The revised manuscript has been significantly improved, both by the additional of new experiments as well as the edits made to the text to better reflect the data presented. Specifically, the new experiments shown in Figure 3A-B and Figure 4J-K examine how the regulation of IR endocytosis by DNAJA2 is functionally related to IR signaling are well conducted. These experiments support the conclusion that DNAJA2 regulation of IR endocytosis modulates IR signaling, which in turn may impact whole body glucose homeostasis.

I have no further comments. This is a strong study that will be of interest to a wide range of researchers in the fields of cell biology, metabolism, and physiology.

Version 2:

Reviewer comments:

Reviewer #1

(Remarks to the Author)

The authors has addressed the questions nicely. I don't have any questions.

REVIEWER COMMENTS

Reviewer #1 (Remarks to the Author):

The authors analyze the effect of DNAJA2, a kind of heat shock protein, on neonatal lethality and glucose homeostasis. They describe a role of DNAJA2 deficiency in facilitating clathrin-mediated endocytosis of InsR. The authors first show that DNAJA2 binds to insulin receptor (InsR). DNAJA2-specific liver knock-out decreased insulin sensitivity and glucose tolerance in chow-fed normal mice. Overall, this is an interesting and important study.

Points need to be addressed:

1. Given that DNAJA2 deficiency (whole body ko and liver-specific ko) had a strong effect on the size of newborn pups (Fig.S1A, and C-D), indicating that these mice have some problems with development and growth. So, the phenotype observed in liver-specific ko mice might be the secondary effect of growth defects. Use the inducible liver-specific ko mice or adenovirus to delete hepatic DNAJA2 when the mice grow up. This is very important.

Thanks for this question. We are sorry for the misleading information. Actually, Fig.S1A, and C-D show only whole-body KO mice, whose growth are affected. Liver-specific KO does not affect the mouse survival, development or growth (Figure S2F). We have the description in the line 115-116: "We observed that CKO mice were born at the normal Mendelian ratio, survived to adulthood (Figure S2F), and were morphologically indistinguishable from WT mice", which rules out the possibility that the observed phenotype is the secondary effect of growth defects.

2. Conclusion of the first paragraph in the discussion section:" Therefore, our findings provide new mechanistic insights into insulin resistance and glucose metabolism, establishing DNAJA2 as a new biomarker for T2DM predisposition and pathogenesis." Right now, the phenotype is observed on chow-fed normal mice. To support this conclusion, they need to evaluate the phenotype of liver DNAJA2 ko on HFD feeding. plus these mice are better to be inducible ko. Additionally, they need to prove that hepatocyte DNAJA2 is downregulated in the insulin-resistant and/or diabetic models (high fat diet fed mice or ob/ob mice or db/db mice) compared to controls.

Thank the reviewer for this question. However, we think this question might be out-of-scope in this manuscript, which can be further pursued in the future studies. In this paper, we clearly shows that DNAJA2 deficiency causes insulin resistance, hyperglycemia and hyperinsulinemia, which are typical characteristics of T2DM. In addition, DNAJA2 expression level is downregulated in islet samples from human T2DM patients and ob/ob mice. Therefore, our data suggests DNAJA2 as a new genetic factor predisposing to T2DM. Lastly, type 2 diabetes is induced by multiple genetic-environmental cross-talks, not necessarily induced by HFD (Nat Rev Endocrinol. 2018;14(2):88-98; J Diabetes Investig. 2023;14(4):503-515; Mol Metab. 2019;27S(Suppl):S139-S146). There are a lot of top papers investigating genetic factors that regulates insulin resistance and glucose homeostasis without using HFD model (Cell. 2016;166(3):567-581; Nature. 2021;590(7845):326-331). We believe that DNAJA2 deficiency is a driver but not HFD-induced passenger for T2DM. To save mice and animal welfare, it is not essentially necessary to perform additional mouse studies.

3. The protein levels of DNAJA2 in the liver and other tissues.

Thanks for the suggestion. We have checked the protein levels of both DNAJA2 and IR β in the brain, liver, lung, heart, kidney, spleen and muscle tissues. As shown in the following figure, IR β shows some tissue-specific expression pattern, but the DNAJA2 expression levels are quite similar among different tissues.

4. The authors analyzed the DNAJA2 expression levels from published datasets in the human samples with/without diabetes. DNAJA2 expression was downregulated in the islets from T2DM patients compared to non-diabetic individuals. However, DNAJA2 was upregulated in the adipocytes of obese mice compared to lean controls (Figure 6E), and a high-sugar-and-high-fat diet (HSD) also induced higher DNAJA2 expression in mouse hepatocytes (GSE18236528) (Figure 6F). The result of the upregulation of DNAJA2 in the hepatocytes contradicts the finding that DNAJA2 deficiency led to insulin resistance (Figure 1). Probably, the protein levels and mRNA levels are not consistent in the case of DNAJA2. It's better to measure the DNAJA2 protein levels in the liver samples of human subjects with diabetes or not.

Thanks for pointing this out. This can be explained as follows: in the case of “DNAJA2 expression was downregulated in the islets from T2DM patients compared to non-diabetic individuals”, DNAJA2 downregulation could be the cause of insulin resistance and T2DM; however, in the case of “a high-sugar-and-high-fat diet (HFD) also induced higher DNAJA2 expression in mouse hepatocytes”, DNAJA2 upregulation might be the result of HFD. As a stress responder, DNAJA2 is upregulated to enhance insulin signaling and to metabolize the excessive glucose. This is the similar case for ob/ob adipose (Figure 6E), which upregulates DNAJA2 to enhance glucose uptake and fat synthesis. To avoid confusion, we deleted Figure 6F in the revised manuscript.

5. The first sentence (Line 68-78) of one paragraph :” The observation that DNAJA2 deficiency induces genomic instability prompted us to investigate its role in spontaneous tumorigenesis in mouse models.” This paragraph just showed the results of the growth and survival. There is no data related to tumorigenesis. Please revise the first sentence of this paragraph.

Thanks for pointing this out. We have added the references (Nat Commun. 2023;14(1):5246; Cell Discov. 2023;9(1):107) here, which are our previous studies showing that DNAJA2 deficiency induces genomic instability, a hallmark of cancer. Based on this observation, we started the mouse study to test tumorigenesis, but got unexpected growth and survival phenotypes.

6. Conclusion of the third paragraph in the results section: “Therefore, our data indicate that insufficient glycogen storage in livers is a major cause of DJ2-/neonatal lethality.” I think this conclusion is not consistent with the finding in DNAJA2-specific liver knock-out mice, which are born at the normal Mendelian ratio and survived to adulthood. In the Discussion Section, the authors then listed another possibility (Line 323-326:” in addition to its function in insulin signaling and glucose metabolism in the liver. For example, DNAJA2 regulates lysosomal pH and lung respiration in mouse models¹¹, which may contribute to the neonatal lethality.”) the authors need to revise the conclusion.

Thanks for this suggestion. We agree with the reviewer that DNAJA2’s functions in other tissues or organs, such as DNAJA2’s function in lung (J Cell Physiol. 2024; 239(2):e31174), may also contribute to the neonatal lethality. Our glucose rescue experiment strongly suggests that DNAJA2’s function in maintaining glucose homeostasis and energy homeostasis is critical for neonatal survival (Figure S2B). Therefore, we have modified the conclusion to “our data indicate that insufficient glycogen storage in livers contributes to DJ2^{-/-} neonatal lethality”.

7. The data about DNAJA2 and InsR interaction is not convincing. (1) There is even a band in the negative control group in the Co-IP experiments (Fig.S5A, S5B). (2) The signal in the PLA experiments is not strong. (3) The co-IP experiments were done in an overexpression system? If so, please repeat the Co-IP experiments to evaluate the endogenous binding in the primary hepatocytes.

Thanks for raising these concerns. The answers are as follows: (1) Heat shock proteins (HSPs), including DNAJA2, mainly interact with client proteins and maintain their proteostasis. Thus, HSPs are relative sticky proteins in Co-IP experiments. In our data, the bands in the IR β Co-IPed group are much stronger than the IgG control group, suggesting that the interaction is convincible. Anyway, we have several repeats and replaced the image with a better gel (Figure S5A). (2) Compared to the negative control, the PLA signal in the experimental group (average 4 foci/cell) is strong and reliable, especially in HepG2 cells with relatively low expression level of DNAJA2. As shown in Figure 5E, the PLA signal is much stronger in 4T1 cells (average 10-15 foci/cell) with a relatively high expression level of DNAJA2. (3) These are all endogenous co-IP experiments, not in overexpression system. In Figure. 4E, we observed IR-DNAJA2 interaction in liver lysates.

8. Does DNAJA2 affect InsR synthesis or turnover? How does pro-InsR look like?

Thanks for these questions. As suggested, we measured InsR protein turnover rate in WT and DJ2^{-/-} cells, and the data (Figures S4C-D) shows that InsR turnover is faster in KO cells than WT cells. This is consistent with the observation that more InsR undergoes spontaneous endocytosis in KO cells, and part of the endocytosed InsR goes for lysosomal degradation.

We have looked at the pro-InsR expression level, and there is no difference between WT and KO cells (Figure 4J and Figure S3D).

9. The data showed that DNAJA2 deficiency promoted InsR endocytosis and location in late lysosome complex (Rab 7 staining results). Generally, in the late lysosome complex, the protein will degrade. As one literature (<https://doi.org/10.1038/s42255-022-00634-5>) showed EphB4 stimulated InsR endocytosis and InsR/Rab7 colocalization in the late lysosome, which resulted in InsR degradation. But the data in Fig.2E and 2F indicate that DNAJA2 deficiency did not affect InsR degradation. Why is that?

Thanks for pointing this out. Actually, in most of our WB results, IR levels are lower in DJ2-KO cells or tissues (Figures 2A, 4A, 4B, Figures S3, S4D, S5A), suggesting partial degradation of this protein in KO cells.

10. I am not quite convinced about the InsR endocytosis upon DNAJA2 deficiency and the authors should show this with an alternative method. i.e. surface biotinylation, that InsR levels at the cell membrane are changed upon DNAJA2 LOF and GOF.

Thanks for the question and suggestion. We have performed membrane fractionation assay. As shown in Figures 3C-D, the InsR protein level on the plasma membrane is significantly reduced in DJ2-KO cells compared to WT cells.

11. Measure the InsR protein level on the cell membrane.

Thanks for the suggestion. The results are shown in Figures 3C-D.

12. It looks to me that DNAJA2 deficiency has a strong effect on the InsR phosphorylation (fig.2A, 2C,2 E, 2F). It's most likely that DNAJA2 deficiency has a direct effect on InsR phosphorylation besides InsR endocytosis.

Thanks for this question. Our data shows that DNAJA2 deficiency impairs membrane localization of InsR, leading to defective InsR response to insulin stimulation (InsR phosphorylation), which is consistent with previous studies showing that spontaneous InsR endocytosis attenuates InsR level on the plasma membrane and inhibits insulin-stimulated InsR phosphorylation (Cell. 2016;166(3):567-581; Nature. 2021;590(7845):326-331).

To further validate that the defective InsR phosphorylation in DJ2-KO cells is caused by InsR endocytosis, we knock out AP2B1, a key factor mediating InsR endocytosis, in DJ2-KO cells. As shown in Figures S5F-G, the membrane localization of InsR is restored, and insulin-stimulated InsR phosphorylation is also restored in the double knockout (DKO) cells (Figures 4J-K), suggesting that DNAJA2 deficiency impairs InsR phosphorylation by promoting spontaneous InsR endocytosis.

Since InsR undergoes autophosphorylation induced by insulin, the reviewer may suggest that DNAJA2 inhibits phosphatase activity against InsR. To test this possibility, we treated WT and DJ2-KO 4T1 cells with a pan-phosphatase inhibitor, and measured insulin-stimulated InsR phosphorylation. As shown in Figure S3D, insulin treatment significantly stimulate InsR phosphorylation in WT cells but not in DJ2-KO cells in the presence of phosphatase inhibitor, suggesting that DNAJA2 does not modulate insulin stimulated InsR phosphorylation through affecting phosphatase activity.

13. The impaired InsR phosphorylation due to DNAJA2 deficiency will lead to insulin resistance in the hepatocytes. Function assay like gluconeogenesis should be performed in primary hepatocytes.

Thanks for this insightful suggestion. We measured glycogen synthase activity of insulin stimulated livers in WT and DJ2-CKO mice. As shown in Figure 1I, the glycogen synthase activity is significantly reduced in CKO mice compared to WT mice, suggesting that DNAJA2 deficiency impairs hepatocyte function.

14. It's nice that the authors discussed other proteins (MAD2, Inceptor), which also facilitate InsR endocytosis. Please include EphB4, which was shown to mediate the effect of insulin on InsR endocytosis.

Thanks for pointing this out. We have included this reference in the revised manuscript.

15. Please indicate age, sex, number of mice in the figure legend. Also, for cell experiments, provide the treatment detail (time, concentration) in the figure legend.

Thanks for these suggestions. We have modified the texts accordingly.

16. For the WB results, annotate them with molecular weight labels, so the readers could have a general idea about the molecular weight of detected proteins.

Thanks for the suggestion. We have modified the figures as suggested.

17. Glucose and insulin tolerance tests in Fig 1G and H are consistent with the overall hypothesis of the DNAJA2-InsR interactions. Since these significant effects are small, the authors attempt to improve the impact of their figures by starting the y-axis some distance from zero. I think it is generally best to start the y-axis at zero, so comparisons are easier for the reader throughout the manuscript.

Thanks for pointing this out. The figures have been modified as suggested.

Reviewer #2 (Remarks to the Author):

This study examines the role of the DNAJA family of chaperones, DNAJA2 (DJ2), in regulation of whole-body insulin signaling and glucose homeostasis, as well as in regulation of insulin action and signaling focusing on the liver. For in vivo studies, two mouse models were examined, a whole-body DJ2 $-/-$ model as well as a liver-specific DJ2 knockout model (CKO). Whole-body DJ2 $-/-$ exhibited neonatal lethality with

largely normal organ development, with the exception of liver abnormalities that included reduced glycogen storage. The neonatal viability phenotype was rescued by subcutaneous injection of glucose, indicating that disruptions of glucose metabolism contributed to the lethality phenotype in DJ2 $-/-$ animals. CKO mice exhibited normal liver development and did not exhibit neonatal loss of viability, but also demonstrated disruptions in glucose metabolism which included elevated blood glucose and insulin levels in the fed state.

IR signaling (focusing on IR beta phosphorylation) was found to be reduced in both DJ2 $-/-$ liver lysates as well as in liver samples of CKO mice 10 min following injection of a pulse of insulin. Reduced pIR β phosphorylation and reduced levels of phosphorylation of other insulin signaling proteins upon disruption of DJ2 was also observed in hepatocytes isolated from DJ2 $-/-$ animals as well as in HepG2 and other cell lines upon silencing DJ2. In HepG2 cells, silencing DJ2 led to a reduction of IR levels at the plasma membrane (relative to cytosol levels) prior to insulin stimulation (basal state) and also reduced the uptake of fluorescently-conjugated insulin. The decrease in PM/C (plasma membrane to cytosol) levels of IR in DJ2 silenced cells was reversed by treatment with the endocytosis inhibitor dynasore or by silencing the clathrin-mediated endocytosis adaptor protein AP2, suggesting that the loss of DJ2 enhanced clathrin-mediated endocytosis of IR. DJ2 silencing was without effect on the cell surface levels of transferrin receptor or transferrin internalization, suggesting that DJ2 specifically regulated clathrin endocytosis of IR but did not broadly impact regulation of all clathrin endocytosis cargo.

DJ2 co-precipitated with IR β , and silencing or overexpression of DJ2 led to an increase or reduction, respectively, of the association of IR β with AP2, detected by co-IP or PLA. The interaction of IR β with DJ2 detected by co-IP or PLA required the C-terminal and tyrosine kinase domains of IR and the J domain of DJ2. Lastly, some analysis of the expression data of various previous studies seems to suggest that DJ2 exhibits alterations in various tissues that correlates with insulin resistant or T2DM conditions.

Overall, this is an interesting study that provides new insights into the regulation of insulin receptor signaling, liver function, and whole-body glucose metabolism. The role of DJ2 in the regulation of insulin receptor membrane traffic provides important mechanistic insights to accompany the *in vivo* work using two different DJ2 knockout

mouse models that establish the physiological relevance of DJ2 in glucose metabolism and insulin action. The interaction experiments using co-IP and PLA are largely well conducted and provide some insights into how DJ2 may regulate IR endocytosis. Moreover, the experiments that dissect the specific regions of DJ2 that interact with IR are very useful to support a model of more direct action of DJ2 on binding to IR to modulate IR endocytosis, rather than considering more indirect effects of DJ2 on insulin signaling. This study will be of interest to a wide range of cell biology and metabolism researchers, but should first look to address the following comments:

We really appreciate the positive comments from this reviewer.

Major comments

1) The experiments included in this study probing the interaction of DJ2 with IR β do not allow conclusion of a direct interaction of these two proteins. Both IP and PLA do not allow resolution of direct interactions, which requires study of interaction of recombinant proteins or other methods. With the data included in this study, it appears that IR β and DJ2 interact, but this cannot conclude direct or indirect interactions. Similarly, the model by which AP2 and DJ2 compete for binding to IR depends on direct interactions of each with IR, which has not been demonstrated by the collective experiments included in this study. Either additional experiments should be included to support the model of direct interaction, or the conclusions and interpretation of these results should be tempered.

Thanks for pointing this out. We agree with the reviewer and have modified the conclusions in the corresponding parts without describing direct interactions as suggested.

2) Lines 59-61 of the introduction state the following conclusion based on the results of the current study: "The spontaneous endocytosis of IR significantly reduces its plasma membrane localization, causing insulin resistance and disrupting glucose metabolism and homeostasis." The causal link between the increased rate of basal endocytosis of IR and insulin resistance and disrupted glucose metabolism (upon DJ2 perturbation) was not established, and conclusions stating as such should thus be revised. Please review the rest of the manuscript for similar statements about causation that are not supported by the data, including for example lines 301-303.

Thanks for pointing this out. Based on our results in the revised manuscript, we have established the causal relationship between DNAJA2 deficiency and increased rate

of basal endocytosis of IR and insulin resistance in vitro (Figures 2-4, also see point #9 below). We have reviewed the whole manuscript and modified inappropriate statements as suggested.

3) Lines 165-176 and Figure 3A-B and S4: Dynasore is not considered a specific inhibitor of clathrin-mediated endocytosis, see PMID: 24046449. The study includes experiments that rely on disruption of clathrin endocytosis with the more selective strategy of silencing AP2, which supports the conclusion that the disrupted membrane traffic of IR is clathrin-endocytosis-dependent, but care should nonetheless be taken when discussing dynasore.

Thanks for this suggestion. We have modified the description in the revised manuscript text.

4) Are these sites in the CTD and the TKD known to bind AP2? What is the evidence and can any previous studies that established this be cited here (lines 224-225)? Does this mutant CTD-TKD of IR-beta also exhibit reduced internalization, for instance upon stimulation with insulin? This additional experiment would be helpful in establishing support for the model that is presented.

Thanks for this constructive suggestion. No paper has directly reported the AP2 binding sites in IR protein. We predicted this according to the reported AP2 binding motifs (Y-X-X-[FILMV] and [ED]-X-X-X-L-[LI]) in the cytosolic tail of transmembrane cargo proteins. Therefore, we identified three putative AP2 binding motifs (named motif 1, 2, 3) in the TKD domain of IR (Figures S6A-B).

To test the effects of these motifs on IR internalization, we first mutated all the three putative motifs, and found that these mutations significantly attenuate pro-IR processing and most of the immature IR are trapped in ER (Figures S6B-C). Next, we tested different mutants with mutations in two motifs or just single one motif, and found that mutating motif 1 did not affect IR processing (Figure S6B), and this mutant inhibits IR-AP2B1 interaction (Figure 5A) as well as IR internalization (Figures 5B-C), suggesting that motif 1 is critical for AP2-mediated IR endocytosis.

5) As an additional extension to the point above, does insulin stimulation trigger a loss of DJ2 interaction with IR? This should be a relatively straightforward experiment to probe (e.g. by co-IP) and would strengthen the conclusion that DJ2 impairs internalization of IR.

Thanks for this constructive suggestion. Our data shows that insulin stimulation does not affect IR-DJ2 interaction (Figures S5B-D), suggesting constitutive interaction between these two proteins, similar to IR-MAD2 interaction as reported previously

(Cell. 2016;166(3):567-581). These data suggest that DNAJA2 prevents spontaneous IR endocytosis under basal condition. However, insulin stimulated IR conformational change can induce IR internalization that is not counteracted by DNAJA2.

6) Lines 241-243: What is meant by an endogenous co-IP experiment? This experiment appears to be done in cell that are DJ2 knockouts that have been rescued with various DJ2 constructs, which cannot be readily considered “endogenous”. Please provide a clearer explanation of what is meant by “endogenous” here.

Thanks for pointing this out. We have deleted “endogenous” here.

7) It is not clear what the rationale for examination of the levels of DJ2 expression in various tissues (Figure 6) may be. Many of the studies examined appear to be in islets (pancreatic islets). However, the role of DJ2 in regulation of endocytosis in islets (and indeed in any of the specific cell types of islets) is not clear and not probed by the current study. Moreover, the role of blood levels of DJ2 has not been examined, and so it is very difficult to understand how examining the blood levels of DJ2 in samples obtained from patients with insulin resistance or T2DM reveals information pertinent to this study. This analysis of patient samples could be removed from the manuscript, or alternatively, careful consideration and justification for study of the expression levels of DJ2 in this broad array of tissue samples should be provided.

Thanks for pointing this out. Our data shows that DNAJA2 regulates IR endocytosis and insulin signaling in various cell lines, including HepG2 (liver), 4T1 (breast), HeLa (cervical) and MC38 (colorectal) cells (Figures 2F, 3, S3, S4), suggesting that DNAJA2 has no tissue specificity in terms of regulating insulin signaling. Based on this, we analyzed available sequencing data that covers DNAJA2 in public database, and determined the association between DNAJA2 expression levels and metabolic features here. To clarify this point, we added justifications in the revised manuscript before we describing these data.

8) Figure 4A, and more importantly Figure 4E show detection of DJ2 with a red arrow. The band indicated is very weak relative to a much stronger band at a higher molecular weight. In Figure 4E, it is expected that DJ2 is absent from the knockout sample, it appears it could be, but the blot shown is not of sufficient clarity and should be replaced.

Thanks for pointing this out. The top band is IgG heavy chain, which is labeled in the revised figures. Because the molecular sizes of DNAJA2 and IgG heavy chain are

very close, they can not be separated very well. We have repeated these co-IP experiments for several times, and the data show reliable interaction between DNAJA2 and IR. We replaced Figure 4E with a new one as suggested. In addition, the IR-DNAJA2 interaction was also confirmed in other cell lines (Figures 4F-G, Figures S4A-C) using a mouse IR antibody for co-IP experiments, which shows clear DNAJA2 band without showing IgG heavy chain.

9) Building on comment 1 above about the lack of support for a causal relationship for the role of DJ2 in regulating IR signaling and endocytosis in liver to whole-body glucose metabolism, there is also a lack of support for a causal relationship between DJ2 regulation of IR endocytosis and IR signaling. The latter can be much more readily probed, for instance in cell lines. It would be very useful to examine how either (i) loss of endocytosis by AP2 silencing (ii) loss of endocytosis of IR specifically with a mutant that cannot bind DJ2 but retains binding to AP2 (if this exists) impacts insulin signaling upon silencing of DJ2. If loss of IR endocytosis (e.g. by loss of AP2) can rescue insulin signaling in DJ2-silenced cells, this would significantly support this aspect of the conclusions and model.

Thanks for the insightful suggestion. We have done the experiments as suggested. Our data shows that depleting AP2B1 in DJ2^{-/-} cells restores the IR protein level on plasma membrane (Figures 3A-B). We further confirmed this in 4T1 cells depleted of AP2B1 (Figures S5F-G). Indeed, AP2B1 deficiency restores the insulin signaling in DJ2^{-/-} cells (Figures 4J-K), suggesting that DNAJA2 regulates insulin signaling through modulating AP2-mediated IR endocytosis.

10) Some consideration of the fate of internalized IR upon DJ2 silencing could be warranted in the discussion. Rab7 should demark late endosomal compartments, such that increased localization of IR to Rab7 compartments upon DJ2 loss of function might be expected to contribute to IR degradation, but the levels of IR do not appear to change. This point could be addressed in the discussion.

Thanks for this suggestion. Indeed, IR level was lower in DJ2^{-/-} cells in most of our WB results (Figures 2A, 4A, 4B, 4J, Figures S3, S4C, S5A). We also performed Cycloheximide (CHX) chase experiment to determine the half-life of IR. The result shows that IR degradation is faster in DJ2^{-/-} cells than WT cells (Figures S4C, D), indicating that DJ2 loss contributes to IR degradation.

Minor comments:

1) Line 639: domians (misspelling)

Thanks for pointing this out. We have corrected the misspelling.

2) Line 152: describing the disruption of IR internalization as “unscheduled” endocytosis of IR is unusual – implies that there is a disruption of the timing of IR in response to a specific cue, while this is perhaps better considered as a disruption of the plasma membrane retention of IR in the basal state.

Thanks for the suggestion. We have changed “unscheduled” to “spontaneous”.

3) Figure 3: the figure caption indicates that panels E-F represent the quantification of internalized insulin and transferrin, but only that of transferrin appears to be shown.

Sorry for the unclear labeling. Panel E shows Cy3-labeled insulin intensity and F (now Figure S4G) shows transferrin intensity.

Reviewer #1 (Remarks to the Author):

The authors addressed most of the questions. With regard to the response to question 1 and question 9, it's still not clear.

1. Given that DNAJA2 deficiency (whole body ko and liver-specific ko) had a strong effect on the size of newborn pups (Fig.S1A, and C-D), indicating that these mice have some problems with development and growth. So, the phenotype observed in liver-specific ko mice might be the secondary effect of growth defects. Use the inducible liver-specific ko mice or adenovirus to delete hepatic DNAJA2 when the mice grow up. This is very important.

Thanks for this question. We are sorry for the misleading information. Actually, Fig.S1A, and C-D show only whole-body KO mice, whose growth are affected. Liver-specific KO does not affect the mouse survival, development or growth (Figure S2F). We have the description in the line 115-116: "We observed that CKO mice were born at the normal Mendelian ratio, survived to adulthood (Figure S2F), and were morphologically indistinguishable from WT mice", which rules out the possibility that the observed phenotype is the secondary effect of growth defects.

Question: Figure S2F shows the survival curves of wild-type (WT), heterozygous (Het), and liver-specific DNAJA2 knockout (CKO) mice. This data does not support the conclusion that the observed phenotype in knockout mice is not a secondary effect of growth defects. Therefore, data similar to that presented in Figures S1C and S1D should be provided.

Thanks for this suggestion, and we have included these data as suggested. As shown in Figures S2G-H, the morphology, size and growth of WT and CKO mice are quite similar, suggesting that the observed phenotype in knockout mice is not a secondary effect of growth defects.

9. The data showed that DNAJA2 deficiency promoted InsR endocytosis and location in late lysosome complex (Rab 7 staining results). Generally, in the late lysosome complex, the protein will degrade. As one literature (<https://doi.org/10.1038/s42255->

022-00634-5) showed EphB4 stimulated InsR endocytosis and InsR/Rab7 colocalization in the late lysosome, which resulted in InsR degradation. But the data in Fig.2E and 2F indicate that DNAJA2 deficiency did not affect InsR degradation. Why is that?

Thanks for pointing this out. Actually, in most of our WB results, IR levels are lower in DJ2-KO cells or tissues (Figures 2A, 4A, 4B, Figures S3, S4D, S5A), suggesting partial degradation of this protein in KO cells.

Question: Please explain why IR remains unchanged in Figures 2E and 2F.

Thanks for the further questions. The hepatocytes used in Figure 2E were isolated from two different mice, the KO cells were not derived from the parental WT cells, the unchanged IR protein level could be resulted from individual difference. As shown in the figure below, the IR protein levels in the same tissue (e.g., heart, kidney, muscle and liver) are actually different among different individuals.

The unchanged IR protein level in Figure 2F could be resulted from another function of DNAJA2 in IR protein turnover. DNAJA2, together with other chaperones (e.g., HSC70), plays important roles in proteolysis pathways, including both autophagy and proteasome degradation pathways (Cell Discov. 9(1):107; Nat Commun.14(1):5246; J Biol Chem. 294(11):4247-4258). Here, we have demonstrated that DNAJA2 interacts with IR. The interaction between IR and DNAJA2 as well as other chaperones was also shown in the BioGRID database. Therefore, it is possible that DNAJA2 is required for IR protein degradation. DNAJA2-deficiency induces IR endocytosis and trafficking to late endosome in HepG2 cells, but may also inhibits IR degradation, which together results in unchanged IR protein level.

Reviewer #2 (Remarks to the Author):

The revised manuscript has been significantly improved, both by the additional of

new experiments as well as the edits made to the text to better reflect the data presented. Specifically, the new experiments shown in Figure 3A-B and Figure 4J-K examine how the regulation of IR endocytosis by DNAJA2 is functionally related to IR signaling are well conducted. These experiments support the conclusion that DNAJA2 regulation of IR endocytosis modulates IR signaling, which in turn may impact whole body glucose homeostasis.

I have no further comments. This is a strong study that will be of interest to a wide range of researchers in the fields of cell biology, metabolism, and physiology.

We really appreciate the positive comments from this reviewer.

Reviewer #1 (Remarks to the Author):

The authors has addressed the questions nicely. I don't have any questions.

Thank the reviewer for taking time to review our paper, and we are glad to have addressed all the questions.